# OpenReview forum: "SlotGCG: Exploiting the Positional Vulnerability in LLMs for Jailbreak Attacks"
_ICLR.cc/2026/Conference — ICLR 2026 Poster_

### Official Review · Reviewer_dH2T · 2025-10-29

**Soundness:** 3
**Presentation:** 3
**Contribution:** 3
**Rating:** 6
**Confidence:** 3

**Summary:**

This paper proposes SlotGCG, which extends GCG jailbreak attacks by inserting adversarial tokens at multiple vulnerable positions throughout prompts rather than only at the suffix. The method uses a Vulnerable Slot Score based on attention patterns to identify optimal insertion positions. Experiments on 6 LLMs show average 14% ASR improvement, faster convergence, and 42% higher robustness against defenses.

**Strengths:**

1. well-motivated problem: The systematic exploration of positional vulnerability is underexplored.

2. Comprehensive empirical validation: Testing across 6 models × 4 attack variants × 4 defenses with consistent improvements demonstrates robustness of the approach.

3. Practical efficiency: The method adds only 200ms preprocessing but achieves up to 10× faster convergence, making it immediately deployable as a drop-in enhancement to existing GCG-based methods.

**Weaknesses:**

1. SlotGCG shows no improvement or degradation on Mistral-7B and Vicuna-7B in Table 1, but the paper provides no analysis of why positional vulnerability varies across architectures. This limits understanding of when the method applies.

2. The observation that defenses can increase ASR  due to GPT-4 filtering during optimization suggests the evaluation methodology itself may be problematic, undermining confidence in the reported improvements.

3. Some hyperparameters lack justification, e.g., why temperature T=8? What happens with other layer selections or temperatures?

**Questions:**

1. Can you characterize what architectural or training differences cause SlotGCG to fail on Mistral/Vicuna?

2. What is the performance with (1) only lower layers, (2) only upper layers, (3) random layer selection?

3. AutoDAN also uses flexible token placement. How does SlotGCG compare in effectiveness and efficiency?

---

> ### Author Response · Authors · 2025-11-21
>
> We appreciate the thoughtful feedback on our paper, and we included additional comments to further improve our work.
>
> > ### **[W1, Q1] SlotGCG performance on Mistral-7B and Vicuna-7B**
> >
>
> Thank you for your comment, which raises the need to analyze why SlotGCG shows limited improvement on Mistral-7B and Vicuna-7B.
> The restricted improvement on these two models is not due to a variation in positional vulnerability across architectures, but rather because the **baseline ASR was already saturated**.
> As has been reported in prior work [1, 2, 3], Mistral-7B and Vicuna-7B are highly susceptible to jailbreak attacks.
> To ensure comparability with prior work, we report **unfiltered results as $ASR_{GPT}$**. In contrast, **manually filtered** results are referred to as **$ASR_{Human}$.**
>
> The revised version will report $ASR_{Human}$ and $ASR_{GPT}$ separately.
>
> **Table A: ASR Comparison on Mistral-7B and Vicuna-7B**
>
> | **Method** | LLaMA2-7B |  | LLaMA2-13B |  | LLaMA3.1-8B |  | Mistral-7B |  | Vicuna-7B |  | Qwen2.5-7B |  |
> | --- | --- | --- | --- | --- | --- | --- | --- | --- | --- | --- | --- | --- |
> |  | $ASR_{Human}$ | $ASR_{GPT}$ | $ASR_{Human}$ | $ASR_{GPT}$ | $ASR_{Human}$ | $ASR_{GPT}$ | $ASR_{Human}$ | $ASR_{GPT}$ | $ASR_{Human}$ | $ASR_{GPT}$ | $ASR_{Human}$ | $ASR_{GPT}$ |
> | **GCG** | 52.00% | 66.00% | 58.00% | 66.00% | 56.00% | 62.00% | 86.00% | 98.00% | 80.00% | 94.00% | 68.00% | 98.00% |
> | **GCG_hij** | 76.00% | 90.00% | 78.00% | 92.00% | 62.00% | 78.00% | 84.00% | 98.00% | 86.00% | 94.00% | 68.00% | 98.00% |
> | **AttnGCG** | 42.00% | 62.00% | 16.00% | 20.00% | 44.00% | 58.00% | 94.00% | 100.00% | 88.00% | 98.00% | 74.00% | 86.00% |
> | **I_GCG** | 62.00% | 88.00% | 56.00% | 60.00% | 46.00% | 60.00% | 90.00% | 100.00% | 96.00% | 96.00% | 82.00% | 100.00% |
> | **GCG + Ours** | 82.00% | 96.00% | 78.00% | 86.00% | 82.00% | 90.00% | 86.00% | 100.00% | 86.00% | 98.00% | 68.00% | 98.00% |
> | **GCG_hij + Ours** | 90.00% | 98.00% | 90.00% | 94.00% | 88.00% | 98.00% | 80.00% | 100.00% | 82.00% | 96.00% | 72.00% | 100.00% |
> | **AttnGCG + Ours** | 92.00% | 94.00% | 92.00% | 94.00% | 80.00% | 88.00% | 90.00% | 100.00% | 90.00% | 94.00% | 88.00% | 98.00% |
> | **I_GCG + Ours** | 90.00% | 98.00% | 94.00% | 96.00% | 72.00% | 92.00% | 90.00% | 100.00% | 96.00% | 98.00% | 86.00% | 96.00% |
>
> As shown in this table, Mistral-7B achieved 98–100% (based on $ASR_{GPT}$) and Vicuna-7B achieved 94–98% (based on $ASR_{GPT}$), demonstrating an ASR close to 100%. This means the baseline performance was already hitting the ceiling, which structurally constrained the potential for SlotGCG to achieve higher ASR.
>
> > **[A for W1, Q1]** Positional vulnerability exists across all architectures, but **baseline $ASR_{GPT}$ already exceeding 94% limits observable improvement on Mistral-7B and Vicuna-7B.**
> >
> ---
> **References**
>
> [1] Zou, Andy, et al. "Universal and transferable adversarial attacks on aligned language models." *arXiv preprint arXiv:2307.15043* (2023).
>
> [2] Huang, Yangsibo, et al. "Catastrophic jailbreak of open-source llms via exploiting generation." *arXiv preprint arXiv:2310.06987* (2023).
>
> [3] Struppek, Lukas, et al. "Exploring the adversarial capabilities of large language models." *arXiv preprint arXiv:2402.09132* (2024).

---

> > ### Comment · Reviewer_dH2T · 2025-11-26
> >
> > The authors address my concerns effectively, I will maintain my score.

---

> > > ### Author Response · Authors · 2025-11-26
> > >
> > > Thank you for your response. We’re pleased that our explanation helped clarify your concerns. If you have any additional questions, we’d be happy to discuss them.

---

> ### Author Response · Authors · 2025-11-21
>
> > ### **[W2] Reliability of the evaluation methodology**
> >
>
> Thank you for your comment, which helps us clarify the reliability of our evaluation methodology.
>
> I. **Reliability and Generalizability of the Evaluation Methodology.**
>
> In most existing works, Jailbreak evaluation is generally performed using two approaches.
>
> - GPT-based evaluation only [4, 5, 6]
> - GPT-based evaluation + Human Judge [7, 8, 9, 10]
>
> The reason for including a human judge is to increase the accuracy of the evaluation, which specifically considers the harmfulness, realism, and level of detail of the generated output.
>
> The evaluation process consisting of (1) keyword-based initial filtering → (2) early stopping using a GPT-4 judge → (3) final verification by human annotators is **widely adopted in many prior works [10, 11, 12], and this evaluation methodology is considered reliable.**
>
> II. **The Reason ASR Increases when Defense Methods are Applied.**
>
> It may appear unexpected that ASR increases when defense methods are applied. However, this does not indicate an issue with the evaluation; rather, it is a phenomenon that can naturally occur in certain cases under Iteration Early Stopping.
>
> During the iterative optimization process, the prompt gradually strengthens its ailbreak capability. **Such irregular cases can occur** around the **transition point where GPT-based Early Stopping and the Human Judge are integrated.**
>
> The increase in ASR when defense methods are applied can be explained by the following two reasons:
>
> - Defense methods generating less detailed outputs, thereby causing weaker Attack Prompts to produce refusal responses.
> - The specificity and realism criteria used by the human judge
>
> **When defense methods are not applied,** GPT-4 often triggers **Early stopping** on **weaker (low-iteration) prompts** because it judges them as harmful, despite their **low specificity and realism**. In this process, **these outputs were corrected to failures** **by the Human Judge**, and consequently the **$ASR_{Human}$** reported in the paper was **lower** when no defense methods were applied.
>
> In contrast, when **defense methods were applied**, **weaker (low-iteration) prompts** failed to generate harmful outputs and thus **did not trigger GPT-4 Early stopping**, while **stronger (high-iteration) prompts did**.
>
> This means that when **no defense methods were applied**, the outputs were indeed harmful but **lacked specificity, leading the human judge to classify some cases as failures**. When **defense methods were applied,** the **stronger high-iteration prompts produced harmful and detailed responses**.
>
> This outcome occurs only in some cases under weaker defense methods and **does not indicate any issue with the reliability of the evaluation**. However, **to avoid such misunderstandings**, we chose to report both **$ASR_{GPT}$ and $ASR_{Human}$.**
>
> > **[A for W2] The evaluation methodology we use is a reliable approach widely adopted in many prior works, and it is entirely plausible that ASR may increase in some cases due to the application of defenses.**
> >
> ---
> **References**
>
> [4] Liu, Yue, et al. "Flipattack: Jailbreak llms via flipping." *arXiv preprint arXiv:2410.02832* (2024).
>
> [5] Wu, Yu-Hang, et al. "Sugar-Coated Poison: Benign Generation Unlocks Jailbreaking." *Findings of the Association for Computational Linguistics: EMNLP 2025*. 2025.
>
> [6] Wu, Tianyi, et al. "Geneshift: Impact of different scenario shift on Jailbreaking LLM." *arXiv preprint arXiv:2504.08104* (2025).
>
> [7] Shu, Dong, et al. "Attackeval: How to evaluate the effectiveness of jailbreak attacking on large language models." *ACM SIGKDD Explorations Newsletter* 27.1 (2025): 10-19.
>
> [8] Kumarappan, Adarsh, and Ananya Mujoo. "Automating Deception: Scalable Multi-Turn LLM Jailbreaks." *First Workshop on Multi-Turn Interactions in Large Language Models*.
>
> [9] Chao, Patrick, et al. "Jailbreakbench: An open robustness benchmark for jailbreaking large language models." *Advances in Neural Information Processing Systems* 37 (2024): 55005-55029.
>
> [10] Jia, Xiaojun, et al. "Improved techniques for optimization-based jailbreaking on large language models." *arXiv preprint arXiv:2405.21018* (2024).
>
> [11] Saiem, Bijoy Ahmed, et al. "Sequentialbreak: Large language models can be fooled by embedding jailbreak prompts into sequential prompt chains." *Proceedings of the 63rd Annual Meeting of the Association for Computational Linguistics (Volume 4: Student Research Workshop)*. 2025.
>
> [12] Zeng, Yifan, et al. "Autodefense: Multi-agent llm defense against jailbreak attacks." *arXiv preprint arXiv:2403.04783* (2024).

---

> ### Author Response · Authors · 2025-11-21
>
> > ### **[W3, Q2] Hyperparameter choices lack justification**
> >
>
> Thank you for your insightful comment on our justification of the hyperparameter choices. We agree that it is important to clearly explain the choices of temperature $T$, layer selection, and after-chat template tokens. To address this, we have conducted additional experiments. We will include detailed analyses in the appendix.
>
> > ### **[W3-1, Q2]** **Rationale for Selecting Upper-Half Layers and After-Chat Template Tokens**
> >
>
> Our decision to target upper-half layers ($\mathcal{L}_{UH}$) and after-chat template tokens($\mathcal{C}$) is grounded in recent jailbreak analysis studies [13, 14].
>
> These works show that adversarial tokens most effectively hijack the model’s attention in higher transformer layers, where semantic features dominate, and that attention from after-chat template tokens is a primary driver of jailbreak success.
> To validate this within our framework, we have conducted ablations using the **SlotGCG (GCG + Ours) method on the Llama 2-7B model** with different layer selections.
>
> **Table B: Ablation on Layer Selection for SlotGCG**
>
> | layer $L$ | $ASR_{Human}$ | $ASR_{GPT}$ |
> | --- | --- | --- |
> | **random** | 78.00% | 96.00% |
> | **lower-half** | 80.00% | 94.00% |
> | **upper-half (ours)** | 82.00% | 96.00% |
>
> The results above confirm that upper-half layers yield the strongest attack success.
>
> > **[A for W3-1, Q2] The consistent strength of the upper-half layer selection empirically validates our VSS definition, which is based on the known attention dynamics critical for jailbreak success.**
> >
> ---
> > ### **[W3-2] Justification for Temperature ($T$=8)**
> >
>
> To justify the effect of temperature $T$, we compared temperature values of **$T$ = 1, 4, 8, 16, 32, and 64**. We conducted ablations using the **SlotGCG (GCG + Ours) method on the Llama 2-7B model**.
>
> **Table C: ASR Variation across Temperature Settings**
>
> | temperature $T$ | $ASR_{Human}$ | $ASR_{GPT}$ |
> | --- | --- | --- |
> | **1** | 60.00% | 90.00% |
> | **4** | 80.00% | 96.00% |
> | **8 (ours)** | 82.00% | 96.00% |
> | **16** | 74.00% | 96.00% |
> | **32** | 72.00% | 98.00% |
> | **64** | 78.00% | 96.00% |
>
> According to Table C, when **$T$ is too high,** the probability spreads out too much, which disrupts important tokens and often causes the **model’s responses to go off-topic.**
>
> We also observed that although **$T$=4** performs reasonably well under the default setting, its **$ASR_{GPT}$ drops by an average of 45.5%** when defenses are applied. However, when **$T$=8, its $ASR_{GPT}$ drops only by an average of 30.5%.**
>
> Across **various environments, $T$=8 consistently produced the most stable and highest ASR**, so we adopted it as our default configuration.
>
> > **[A for W3-2]** We conducted ablations across a wide range of temperatures (1–64) and found that **T=8 consistently provides the best balance between stability and ASR performance.**
> >
> ---
> **References**
>
> [13] Ben-Tov, Matan, Mor Geva, and Mahmood Sharif. "Universal Jailbreak Suffixes Are Strong Attention Hijackers." *arXiv preprint arXiv:2506.12880* (2025).
>
> [14] Wang, Zijun, et al. "AttnGCG: Enhancing jailbreaking attacks on LLMs with attention manipulation." *arXiv preprint arXiv:2410.09040* (2024).

---

> ### Author Response · Authors · 2025-11-21
>
> > ### **[Q3] Comparison with AutoDAN**
> >
>
> AutoDAN [15] also performs flexible token placement. Both AutoDAN and SlotGCG aim to move beyond fixed-suffix attacks, but AutoDAN achieves flexibility through segment-level genetic crossover, which **limits its ability to place tokens precisely at the most vulnerable locations.**
>
> In contrast, SlotGCG evaluates **all $L+1$ token-level insertion slots** using the Vulnerable Slot Score (VSS), enabling fine-grained positional targeting that AutoDAN’s crossover-based mechanism cannot realize. Moreover, VSS allows our method to **quantitatively measure how vulnerable each slot is to jailbreak attacks**, whereas **AutoDAN cannot assess** positional **vulnerability** at all.
>
> In terms of **computation**, AutoDAN’s genetic search incurs heavy overhead due to repeated crossover and mutation cycles. SlotGCG identifies vulnerable slots with a single ~200 ms VSS probing step and restricts optimization to high-VSS positions, reducing optimization iterations by 40–60% and achieving up to 10× faster convergence.
>
> In terms of **performance**, prior work shows AutoDAN generally underperforms GCG by more than 10% on standard benchmarks [14, 16]. These results illustrate that **SlotGCG’s token-level positional modeling** yields **substantially higher effectiveness than genetic insertion** based approaches such as AutoDAN.
>
> > **[A for Q3] SlotGCG achieves higher effectiveness and greater efficiency by leveraging token-level slot discovery, surpassing AutoDAN’s genetic placement strategy.**
> >
> ---
> **References**
>
> [14] Wang, Zijun, et al. "AttnGCG: Enhancing jailbreaking attacks on LLMs with attention manipulation." *arXiv preprint arXiv:2410.09040* (2024).
>
> [15] Liu, Xiaogeng, et al. "Autodan: Generating stealthy jailbreak prompts on aligned large language models." *arXiv preprint arXiv:2310.04451* (2023).
>
> [16] Mazeika, Mantas, et al. "Harmbench: A standardized evaluation framework for automated red teaming and robust refusal." *arXiv preprint arXiv:2402.04249* (2024).

---

### Official Review · Reviewer_99aE · 2025-10-30

**Soundness:** 3
**Presentation:** 4
**Contribution:** 3
**Rating:** 6
**Confidence:** 4

**Summary:**

This paper argues that jailbreaking susceptibility depends strongly on where adversarial tokens are inserted. It introduces a Vulnerable Slot Score (VSS) to rank token positions by “positional vulnerability,” and proposes SlotGCG, which allocates/optimizes adversarial tokens at high-VSS slots rather than only at the suffix. Across Llama-2/3-8B, Mistral-7B, Vicuna-7B, and Qwen-2.5, SlotGCG reportedly improves attack success rate (ASR) over several GCG-family baselines, converges in fewer iterations, and remains more effective against several input-filtering defenses.

**Strengths:**

* The paper presents a clear and original problem framing, defining insertion slots, formalizing the Vulnerable Slot Score (VSS), and linking it to attention patterns to show that positional vulnerability is largely prompt dependent.
* The method is attack-agnostic and simple, with a clear step-by-step presentation. The analysis is insightful, using random multi-position insertion and attention heatmaps that convincingly support the positional hypothesis.
* The results show significant empirical gains, with large performance improvements and meaningful reductions in the number of optimization steps needed for success.
* The attack appears more robust than prior methods against several defenses.
* The writing is clear, well-structured, and easy to follow.

**Weaknesses:**

1. The paper lacks an analysis of transferability across models. It remains unclear whether positional vulnerabilities are model-specific or primarily prompt-dependent. Evaluating SlotGCG as a black-box attack would provide valuable insight into this question.
2. The evaluation is limited to AdvBench, while several newer jailbreak or safety datasets now exist [1-3]. Including additional benchmarks would strengthen the empirical claims and demonstrate broader robustness.
3. The method is only tested within the GCG family. Other optimization-based attacks exist, and it is unclear whether the proposed position-finding process generalizes to them. Since the abstract claims applicability to “any” attack, evidence from beyond GCG is needed.
4. The defense selection is weak. The Erase-and-Check (suffix) version is expected to perform poorly when there is no suffix, as it effectively just deletes the response. Evaluating only the SmoothLLM swap defense is also insufficient; since the attack produces more uniform attention maps, token swapping may be less effective. Other SmoothLLM variants (insert, patch) and stronger recent defenses [4] should be tested for a fair assessment.
5. The reported preprocessing cost of “+200 ms” is not empirically demonstrated or discussed in detail. The paper should clarify how this value was obtained.
6. Although the calculation of the Vulnerable Slot Score (VSS) is novel, the general idea that attack performance depends on token position has been explored up to some level in prior work [5-7]. These earlier studies should be acknowledged and discussed.

***Minor remarks:***
1. Table 5 presents very strong and important results, so it would be better placed in the main text rather than the appendix to highlight its contribution more clearly.
2. Line 472 refers to “Table 3” for the VSS distribution, but this table is no related. This reference should be corrected.

[1] Zeng, Y., Shen, T., Ding, Y., Zheng, L., Sun, Y., & Chen, H. (2024). JailbreakBench: An Open Robustness Benchmark for Jailbreaking Large Language Models

[2] Mazeika, M., Wei, A., Casper, S., Rafailov, R., Dragan, A. D., Finn, C., & Hadfield-Menell, D. (2024). HarmBench: A Standardized Evaluation Framework for Automated Red Teaming and Robust Refusal

[3] Xu, W., Wang, X., Zhang, Z., & Li, M. (2023). ToxicChat: Unveiling Hidden Challenges of Toxicity Detection in Real-World User-AI Conversation

[4] Yi, S., Liu, Y., Sun, Z., Cong, T., He, X., Song, J., Xu, K., & Li, Q. (2024). Jailbreak Attacks and Defenses Against Large Language Models: A Survey

[5] Wang, J., Li, H., Peng, H., Zeng, Z., Wang, Z., Du, H., & Yu, Z. (2025). Activation-Guided Local Editing for Jailbreaking Attacks.

[6] Mu, J., Ying, Z., Fan, Z., Jing, Z., Zhang, Y., Yu, Z., & Zhang, X. (2025). Mask-GCG: Are All Tokens in Adversarial Suffixes Necessary for Jailbreak Attacks?

[7] Rocamora, E., Dubey, A., Jauhri, A., Pandey, A., Letman, A., Mathur, A., & Vaughan, A. (2024). Revisiting Character-Level Adversarial Attacks for Large Language Models.

**Questions:**

1. Are token budgets (total adversarial tokens) matched across baselines in Table 1?
2. How sensitive are results to VSS temperature, and number of slots selected?

---

> ### Author Response · Authors · 2025-11-21
>
> We appreciate the thoughtful feedback. We included additional comments to further improve our work.
>
> > ### **[W1] Lacks an analysis of transferability**
> >
> Thank you for the comment. To investigate this aspect, we extended the Universal Prompt Optimization procedure from GCG [1] and designed a surrogate method applicable to SlotGCG. We applied this **surrogate transferability algorithm** in our experiments, and the **complete procedure is provided in Appendix K.**
> The Universal Prompt Optimization procedure in SlotGCG is as follows.
>
> I. **Universal Vulnerable Slot Positions.**
>
> We use the **AggregationSlots** algorithm to locate slots that are **consistently vulnerable across behaviors** by aligning and integrating **per behavior scores into a unified vulnerability representation.**
>
> We begin by computing VSS for each behavior independently. Since behaviors $x^{(j)}$ have different lengths, each per behavior score sequence $\mathrm{VSS}^{(j)}_s$ is linearly interpolated onto a shared global coordinate system. This global system is defined over the universal slot set
>
> $$
> S^{\mathrm{univ}} = \lbrace 0,\dots,L_{\max} \rbrace,
> $$
>
> where $⁡L_{\max}$ is the maximum prompt length across all behaviors.
>
> After interpolation, the scores from all behaviors are averaged position-wise. This produces the **universal Vulnerable Slot Score,**
> $\mathrm{VSS}^{\mathrm{univ}}_{s},$
> representing slot positions that are consistently vulnerable across the entire behavior set.
>
> II. **Mapping Universal Slots to Individual Behaviors.**
>
> To apply universal slot positions to any given behavior, we use the **AttackInput** algorithm. Each universal slot index is rescaled to match the actual length $L_j$ of behavior $j$. The behavior-specific slot position is computed as
>
> $$
> s_t^{(j)} = \mathrm{round}\left(s_t^{\mathrm{univ}} \cdot \frac{L_j}{L_{\max}}\right).
> $$
>
> These mapped positions determine where adversarial tokens would be inserted for behavior $j$. The constructed input,
>
> $$
> \mathcal{I} \big(x^{(j)}, a_{1:\ell}, S^{(j)}_A\big),
> $$
>
> preserves the overarching universal slot structure while adapting it to prompts of differing lengths. This enables consistent application across heterogeneous behaviors.
>
> III. **Universal SlotGCG Optimization.**
>
> We integrate this universal slot mechanism into the universal prompt optimization algorithm introduced in GCG [1]. As new behaviors are added to the optimization set, we recompute $\mathrm{VSS}^{\mathrm{univ}}_{s}$ using the **AggregationSlots** algorithm. The updated universal slots, together with the AttackInput algorithm, are then used to construct the adversarial inputs for each prompt. This provides a unified slot selection strategy that generalizes across prompts with varying lengths and structures. **A full description of the procedure and the corresponding algorithms is provided in Appendix K.**
>
> We optimize the universal adversarial tokens and the universal VSS on Vicuna-7B using the 50 AdvBench behaviors from Sec. 5 over 500 optimization steps. For the 388 transfer behaviors introduced in [1], we constructed attack prompts by inserting **universal adversarial tokens into these universal slots** **using the AttackInput** algorithm. We then evaluated transfer performance on GPT-3.5-turbo, GPT-4o, Gemini 2.0 Flash, Gemini 2.5 Pro, and Vicuna-7B-1.5v. For ASR evaluation, we used only the GPT-based judge (excluding human evaluation), and we denote these scores as **$ASR_{GPT}$.** When manual filtering is included, we refer to the resulting scores as **$ASR_{Human}$**.
>
> **Table A: Transfer $ASR_{GPT}$ on 388 Harmful Behaviors**
>
> | **Method** | **GPT-3.5-turbo** | **GPT-4o** | **Gemini 2.0 Flash** | **Gemini 2.5 Pro** | **Vicuna 7B 1.5v** |
> | --- | --- | --- | --- | --- | --- |
> | **GCG** | 3.09% | 0.00% | 1.29% | 0.00% | 70.10% |
> | **GCG + Ours** | 50.77% | 1.80% | 2.06% | 3.61% | 63.40% |
> | **AttnGCG** | 25.52% | 0.52% | 0.26% | 0.26% | 64.43% |
> | **AttnGCG + Ours** | 43.04% | 1.55% | 0.00% | 0.00% | 76.80% |
> | **I-GCG** | 12.89% | 0.00% | 0.00% | 0.52% | 81.19% |
> | **I-GCG + Ours** | 40.46% | 0.77% | 0.77% | 0.26% | 79.90% |
> | **GCG-hij** | 37.89% | 0.00% | 0.00% | 1.55% | 63.66% |
> | **GCG-hij + Ours** | 41.49% | 0.26% | 4.12% | 6.70% | 61.34% |
>
> From the results above, we examined the transferability of SlotGCG and found that our method **exhibits context robustness, as illustrated by the Vicuna results.** Furthermore, under **cross-model transfer,** **SlotGCG consistently improved** the **transferability** of almost all baseline methods across all evaluated models, with **the average ASR rising from 5.24% to 12.35%.**
>
> > **[A for W1]** We have **added the Universal SlotGCG Optimization to Appendix K**, which **enables transferability experiments**, and we demonstrate that **our method improves cross-model transferability**.
> >
> ---
> **References**
>
> [1] Zou, Andy, et al. "Universal and transferable adversarial attacks on aligned language models." *arXiv preprint arXiv:2307.15043* (2023).

---

> ### Author Response · Authors · 2025-11-21
>
> > ### **[W2] Limited Benchmark Scope in the Evaluation**
> >
>
> Thank you for your helpful comment about enhancing the robustness of our results with more recent jailbreak and safety benchmarks.
> Following your recommendation, we conducted additional evaluations on **HarmBench** and **JailbreakBench** by randomly sampling 50 prompts from each dataset and comparing GCG with GCG + Ours under the same evaluation settings.
>
> **Table B: Additional Jailbreak Benchmark Results**
>
> | Method | HarmBench | JailbreakBench |
> | --- | --- | --- |
> | **GCG** | 82.00% | 82.00% |
> | **GCG + Ours** | 100.00% | 96.00% |
>
> Across all datasets, our method shows **overall improvements of 14–18 percentage points**, and in some cases even larger gains.
> These supplementary results indicate that our method provides consistent and substantial improvements over baseline GCG across **both newer safety benchmarks and the original AdvBench**. We believe this helps strengthen the empirical evidence for the robustness of our approach.
> **The revised version will include these additional results, with full details provided in appendix.**
>
> > **[A for W2]** We **evaluated SlotGCG** on **newer jailbreak and safety datasets**, and it **continues to enhance existing methodologies.**
> >
> ---
> > ### **[W3] Tested only on GCG-based methods.**
> >
>
> Thank you for your comment, which raises an important point regarding the generality of our method beyond the GCG family. We appreciate the opportunity to clarify this aspect.
>
> Our position-finding strategy is compatible with **any gradient-based attacks** that optimize only the inserted adversarial tokens while keeping the rest of the prompt fixed. Since our method operates solely through **slot selection and mapping**, it can be naturally integrated into these attacks without requiring any changes to their optimization procedures.
>
> To demonstrate generality beyond GCG, we also applied our method to **GBDA** [3]. Unlike the discrete token-level optimization used in the GCG family, GBDA updates continuous embeddings and later discretizes them into adversarial prompts.
>
> **Table C: $ASR$ of GBDA with and without Our Method**
>
> | **Method** | **Llama2-7B** |  | **Llama2-13B** |  | **Llama3.1-8B** |  | **Mistral-7B-Instruct-0.2** |  | **Vicuna-7B** |  | **Qwen2.5** |  |
> | --- | --- | --- | --- | --- | --- | --- | --- | --- | --- | --- | --- | --- |
> |  | $ASR_{Human}$ | $ASR_{GPT}$ | $ASR_{Human}$ | $ASR_{GPT}$ | $ASR_{Human}$ | $ASR_{GPT}$ | $ASR_{Human}$ | $ASR_{GPT}$ | $ASR_{Human}$ | $ASR_{GPT}$ | $ASR_{Human}$ | $ASR_{GPT}$ |
> | **GBDA** | 6.00% | 6.00% | 0.00% | 0.00% | 22.00% | 22.00% | 70.00% | 74.00% | 10.00% | 10.00% | 14.00% | 14.00% |
> | **GBDA+Ours** | 44.00% | 66.00% | 4.00% | 6.00% | 66.00% | 92.00% | 74.00% | 98.00% | 18.00% | 24.00% | 34.00% | 84.00% |
>
> Across all models, our method yields a average improvement in GBDA’s ASR, typically increasing success rates from 21.00% to 61.67%. This shows that our method enhances attack effectiveness broadly across models.
>
> Additionally, your comment made us realize that the term “*optimization attacks*” may be interpreted too broadly. **To ensure clarity, we now use the more precise term “gradient-based attacks” in the revised version, following prior work [2].**
>
> > **[A for W3]** By **applying our method** to **GBDA, an optimization-based approach outside the GCG family,** we demonstrated that it can be **easily extended to other optimization-based methods** and **confirmed its effectiveness.**
> >
> ---
> **Reference**
>
> [2] Yi, Sibo, et al. "Jailbreak attacks and defenses against large language models: A survey." *arXiv preprint arXiv:2407.04295* (2024).
>
> [3] Guo, Chuan, et al. "Gradient-based adversarial attacks against text transformers." *arXiv preprint arXiv:2104.13733* (2021).

---

> ### Author Response · Authors · 2025-11-21
>
> > ### **[W4] Concern regarding the strength of defense selection.**
> >
>
> Thank you for your comment, which raises an important point about evaluating our method against defenses. We agree that testing our approach with a broader range of SmoothLLM perturbation types and recent defenses would enhance the reliability of our results.
>
> Following your suggestion, we conducted additional experiments on SmoothLLM variants (Insert, Patch) and several recent defense mechanisms, including RPO [4], SafeDecoding [5], and Llama Guard 3 [6].
> We used the same hyperparameters as **SWAP** for all variations of **SmoothLLM** (6 perturbed variants, $q$ = 5%). **SafeDecoding** was also used with its default settings ($\alpha$ = 3, $m$ = 2, top-$k$ = 10, $c$ = 5). When using **Llama Guard** as a defense, we proceeded with the GPT-4 Judge only when Llama Guard classified the output as safe.
>
> **Table D: Additional Defense Results**
>
> | Method | SmoothLLM (Swap) |  | SmoothLLM (Insert) |  | SmoothLLM (Patch) |  | RPO |  | SafeDecoding |  | Llama Guard 3 |  |
> | --- | --- | --- | --- | --- | --- | --- | --- | --- | --- | --- | --- | --- |
> |  | $ASR_{Human}$ | $ASR_{GPT}$ | $ASR_{Human}$ | $ASR_{GPT}$ | $ASR_{Human}$ | $ASR_{GPT}$ | $ASR_{Human}$ | $ASR_{GPT}$ | $ASR_{Human}$ | $ASR_{GPT}$ | $ASR_{Human}$ | $ASR_{GPT}$ |
> | **GCG** | 44.00% | 44.00% | 22.00% | 38.00% | 24.00% | 62.00% | 32.00% | 46.00% | 8.00% | 10.00% | 16.00% | 26.00% |
> | **GCG + Ours** | 86.00% | 96.00% | 76.00% | 92.00% | 76.00% | 96.00% | 30.00% | 46.00% | 10.00% | 12.00% | 16.00% | 26.00% |
> | **AttnGCG** | 30.00% | 32.00% | 18.00% | 32.00% | 28.00% | 48.00% | 34.00% | 52.00% | 6.00% | 8.00% | 10.00% | 18.00% |
> | **AttnGCG + Ours** | 92.00% | 94.00% | 72.00% | 94.00% | 72.00% | 96.00% | 44.00% | 64.00% | 20.00% | 26.00% | 12.00% | 22.00% |
> | **GCG-Hij** | 44.00% | 44.00% | 32.00% | 54.00% | 52.00% | 78.00% | 42.00% | 54.00% | 8.00% | 8.00% | 16.00% | 26.00% |
> | **GCG-Hij + Ours** | 96.00% | 96.00% | 66.00% | 94.00% | 64.00% | 98.00% | 38.00% | 66.00% | 26.00% | 28.00% | 24.00% | 30.00% |
> | **I-GCG** | 44.00% | 48.00% | 28.00% | 44.00% | 36.00% | 64.00% | 36.00% | 60.00% | 14.00% | 14.00% | 14.00% | 24.00% |
> | **I-GCG + Ours** | 96.00% | 98.00% | 82.00% | 98.00% | 80.00% | 98.00% | 38.00% | 60.00% | 18.00% | 20.00% | 20.00% | 22.00% |
>
> As shown in Table D, SlotGCG also works effectively under other SmoothLLM variants such as **Insert** and **Patch**. On average, the $ASR_{GPT}$ increases from **52.50% to 95.75%**, indicating that our method is not limited to a specific SmoothLLM perturbation type but generalizes well across different variants.
>
> Furthermore, SlotGCG achieves significantly higher Attack Success Rates (ASR) compared to the baseline under recent defenses such as RPO, SafeDecoding, and Llama Guard 3.
>  This demonstrates that **our attack remains effective even against the latest safety alignment techniques. These additional defense results have also been incorporated into the revised version as Table 3.**
>
> > **[A for W4]** **SlotGCG** shows consistently **strong performance across all SmoothLLM variants**, including Insert and Patch. Moreover, **SlotGCG** shows consistently **higher ASR than all baselines across every evaluated stronger defense**, such as RPO, SafeDecoding, and Llama Guard 3.
> >
> ---
> **References**
>
> [4] Zhou, Andy, Bo Li, and Haohan Wang. "Robust prompt optimization for defending language models against jailbreaking attacks." *Advances in Neural Information Processing Systems* 37 (2024): 40184-40211.
>
> [5] Xu, Zhangchen, et al. "Safedecoding: Defending against jailbreak attacks via safety-aware decoding." *arXiv preprint arXiv:2402.08983* (2024).
>
> [6] Inan, Hakan, et al. "Llama guard: Llm-based input-output safeguard for human-ai conversations." *arXiv preprint arXiv:2312.06674* (2023).

---

> ### Author Response · Authors · 2025-11-21
>
> > ### **[W5] Clarification of the preprocessing cost (+200 ms)**
> >
>
> Thank you for your comment, which highlights the need for a more detailed breakdown of the computational overhead. We agree that our reported preprocessing cost of “+200 ms” should be supported with a detailed explanation.
>
> The **only additional overhead comes from Steps 1–3** in Sec. 4, which are performed **once before optimization** and **add no extra operations during the optimization loop.**
>
> Specifically, this overhead consists of **two insertion operations (in Steps 1 and 3)** and a **single inference pass (in Step 2)**. The subsequent optimization process follows the standard GCG implementation (https://github.com/llm-attacks/llm-attacks). With efficient tensor slicing, the optimization stage can keep track of the adversarial tokens, **allowing the optimization loop to run without any additional cost.**
>
> We empirically measured this initialization cost on an NVIDIA A100 GPU, confirming it takes approximately 200ms.
>
> **Table E: Comparison of Preprocessing Time**
>
> |  | GCG | GCG+ours |
> | --- | --- | --- |
> | **Preprocessing time** | N/A | 222.89ms |
>
> Table E summarizes the preprocessing time. In the revised version, we will add this explanation of the preprocessing cost to Sec. 5.4.
>
> > **[A for W5]** The reported “+200 ms” reflects a **lightweight preprocessing step (two insertions and one inference)**, and theoretically, **no additional cost is incurred during optimization.**
> >
> ---
> > ### **[W6] The general idea that attack performance depends on token position**
> >
>
> Thank you for your comment. We appreciate you bringing these relevant studies [7, 8, 9] to our attention. We agree that discussing prior work on positional dependence is crucial for properly contextualizing our contributions. As suggested, we have updated the Related Work section (Appendix A) to explicitly discuss these studies.
>
> > **[A for W6**] We acknowledge **these prior studies on positional effects** and have **explicitly incorporated them into the Related Work section** (Appendix A) to clarify how our VSS formulation builds upon and extends this line of work.
> >
> ---
> > ### **[Minor remarks 1, 2]**
> >
>
> Thank you for your comment and for carefully reading our paper and providing constructive suggestions to improve its clarity and presentation.
>
> > **[A for Minor remarks 1]** We agree that the fast convergence (10× speedup) demonstrated in Table 5 and the significantly reduced number of optimization iterations represent a key contribution of SlotGCG. As you suggested, we have highlighted this result more clearly by moving Appendix K to Sec. 5.4 in the revised version.
> >
>
> > **[A for Minor remarks 2]** Thank you for pointing out the mistake. As you correctly noted, referencing Table 3 in relation to the VSS distribution was an error. We have corrected this citation and replaced it with the appropriate reference to Figure 7, which properly presents the VSS distribution and corresponding attention patterns.
> >
> ---
> **References**
>
> [7] Wang, Jiecong, et al. "Activation-Guided Local Editing for Jailbreaking Attacks." *arXiv preprint arXiv:2508.00555* (2025).
>
> [8] Mu, Junjie, et al. "Mask-GCG: Are All Tokens in Adversarial Suffixes Necessary for Jailbreak Attacks?." *arXiv preprint arXiv:2509.06350* (2025).
>
> [9] Rocamora, Elias Abad, et al. "Revisiting character-level adversarial attacks for language models." *arXiv preprint arXiv:2405.04346* (2024).

---

> ### Author Response · Authors · 2025-11-21
>
> >### **[Q1]  Token budgets matched across baselines?**
> >
>
> All results in Table 1 **use the same adversarial token budget of 20 tokens**, following the standard configuration of the GCG [1]. This ensures that comparisons comparisons across baseline methods and SlotGCG are conducted under a matched optimization setting.
>
> To confirm the stability of our findings, we also conducted additional experiments on Llama-2-7B using different token budgets ($m$=5, 10, 15). A smaller token budget consistently resulted in **lower ASR** and **slower convergence**, whereas larger budgets yielded higher ASR and faster convergence.
>
> **Table F: Effect of Token Budget ($m$) on ASR and Convergence**
>
> | Method | $m$=5 |  | $m$=10 |  | $m$=20 |  |
> | --- | --- | --- | --- | --- | --- | --- |
> |  | $ASR_{Human}$ | $ASR_{GPT}$ | $ASR_{Human}$ | $ASR_{GPT}$ | $ASR_{Human}$ | $ASR_{GPT}$ |
> | **GCG** | 8.00% | 20.00% | 34.00% | 58.00% | 52.00% | 66.00% |
> | **GCG + Ours** | 62.00% | 76.00% | 76.00% | 92.00% | 82.00% | 96.00% |
>
> **Table G: Effect of Token Budget ($m$) on Average Convergence Speed**
>
> | Method | $m$=5 | $m$=10 | $m$=20 |
> | --- | --- | --- | --- |
> | **GCG** | 98.80 iter | 143.00 iter | 138.10 iter |
> | **GCG + Ours** | 68.05 iter | 32.04 iter | 40.50 iter |
>
> These results show that **reducing the token budget $m$** lowers ASR and generally slows convergence, while larger budgets tend to improve both metrics.
>
> For example, the $ASR_{GPT}$ of GCG **increases** from 20% at $m$=5 to 66% at $m$=20, and **our method shows a similar trend**, rising from 76% to 96%.
>
> > **[A for Q1]** All baselines in **Table 1 use the same 20 adversarial token budget**, and our additional experiments show that smaller token budgets lead to reduced ASR and slower convergence.
> >
> ---
> > ### **[Q2] Sensitivity to VSS temperature, and number of slots**
> >
>
> In response, we evaluated the **effect of the temperature by comparing $T$ = 1, 4, 8, 16, 32, 64.**
>
> **Table H:**  **ASR Variation across Temperature Settings**
>
> | **temperature $T$** | **$ASR_{Human}$** | **$ASR_{GPT}$** |
> | --- | --- | --- |
> | **1** | 60.00% | 90.00% |
> | **4**| 80.00% | 96.00% |
> | **8 (Ours)** | 82.00% | 96.00% |
> | **16** | 74.00% | 96.00% |
> | **32** | 72.00% | 98.00% |
> | **64** | 78.00% | 96.00% |
>
> As shown in Table H, when **$T$ is too high,** the probability spreads out too much, which **disrupts important tokens** and often causes the **model’s responses to go off-topic**.
>
> We also observed that although $T$=4 performs reasonably well under the default setting, its **$ASR_{GPT}$ drops by an average of 45.5%** when **defenses are applied.** However, when **$T$=8**, its **$ASR_{GPT}$ drops only** by an average of **30.5%.**
>
> Overall, $T$=8 provides the best balance between stability and effectiveness, and we use it as our default setting.
>
> **Table I:  Temperature Sensitive Selection of Slots and Tokens per-slot**
>
> | **temperature $T$** | **Number of Slots** | **Std. of Slot Count** | **Avg. Tokens per Slot** | **Std. of Tokens per Slot** |
> | --- | --- | --- | --- | --- |
> | **1** | 1.98 | 0.79 | 12.10 | 5.36 |
> | **4** | 7.90 | 2.43 | 2.90 | 1.35 |
> | **8** | 13.82 | 2.34 | 1.50 | 0.32 |
> | **16** | 15.94 | 2.85 | 1.30 | 0.27 |
> | **32** | 16.04 | 2.95 | 1.29 | 0.27 |
> | **64** | 16.04 | 2.95 | 1.29 | 0.27 |
>
> Table I shows that **higher temperatures select more slots** (from 1.98 to 16.04) with **fewer tokens per slot** (from 12.10 to 1.29), while **lower temperatures** produce **greater variation in token counts per slot** (from 5.36 to 0.27).
>
> > **[A for Q2] Higher temperatures select more slots** with **fewer tokens per slot**, which can **increase** coverage but may **hurt ASR by causing off-topic generations**. **Lower temperatures** produce fewer slots that can lead to **larger ASR drops under defenses.**
> >
> ---
> **References**
>
> [1] Zou, Andy, et al. "Universal and transferable adversarial attacks on aligned language models." *arXiv preprint arXiv:2307.15043* (2023).

---

> > ### Comment · Reviewer_99aE · 2025-11-27
> >
> > The authors have addressed most of my concerns. I will increase my score accordingly.

---

> > > ### Author Response · Authors · 2025-11-27
> > >
> > > Thank you for your response and for raising your rating. We're pleased that our rebuttal was able to address your concerns. Your insight would be very helpful to improve our paper.

---

### Official Review · Reviewer_rRJL · 2025-10-31

**Soundness:** 3
**Presentation:** 2
**Contribution:** 3
**Rating:** 4
**Confidence:** 3

**Summary:**

The paper introduces SlotGCG, a positional variant of GCG that exploits positional vulnerability in LLMs. Instead of appending adversarial tokens as a suffix, SlotGCG identifies vulnerable token slots within the prompt using a lightweight Vulnerable Slot Score (VSS) derived from attention patterns, then inserts and optimizes attack tokens at those positions. The method is attack-agnostic and can be used as a plug-in front end to multiple GCG-style optimizers with minimal extra overhead. Experiments on several open-source models report higher ASR, faster convergence, and improved robustness under certain defenses, with success judged via automatic and human checks.

**Strengths:**

1. Novelty

Reframes jailbreak optimization from “suffix-only” to positional attacks by identifying vulnerable token slots via a lightweight attention-derived score (VSS) and inserting/optimizing adversarial tokens at those positions.


2. Method is general, plug-and-play, and more efficient


Attack-agnostic front end that can be attached to multiple GCG-based optimizers with minimal overhead.
Results show faster convergence/fewer steps and higher ASR than standard suffix-only pipelines under comparable budgets.

3. Good experimental coverage

Evaluates across several commonly used open-source instruction models (e.g., Llama, Mistral, Vicuna, Qwen).
Adapts to multiple GCG-based attack variants and compares under several defenses, demonstrating consistent gains.

**Weaknesses:**

1. Threat model and usability boundaries

The core VSS metric depends on attention weights (upper-half layers from the after-chat template to adversarial tokens), which are typically unavailable in black-box/closed models. The paper does not clarify applicability in strict black-box settings or provide surrogate attack choice.

2. Transferability is underexplored
  - Cross-model transfer: Do attack prompts found on one model transfer to other models without further optimization (zero-shot transfer)?
  - Seed sensitivity: How does optimization vary with different random seeds (initial tokens, sampling orders)?
  - Context/system-prompt robustness: For the same target model, does changing the system prompt or different context affect ASR?

3. Recency of attack targets

Experiments focus on open-source instruction models (the newest being Qwen-2.5). There is no demonstration on newer/stronger/closed-source LLMs, limiting external validity.

4. Hyperparameter choices lack justification

The effects of temperature in VSS, the precise definition of “upper-half layers,” and the impact of different after-chat template tokens are not detailed and analyzed. It remains unclear how sensitive VSS and final ASR are to these design choices.

5. Confusion in section THE ROBUSTNESS OF SLOTGCG UNDER DEFENSE METHODS

Perplexity Filter yields 0 ASR for all attack variants, yet the paper claims “Erase-and-Check yields the largest reduction in ASR.” This seems to appear inconsistent.

The paper attributes some failures to GPT-4 misclassification due to biases in the GPT-based filtering mechanism, but overall ASR is still measured by the same GPT-based judge. This creates a tension: if the judge is unreliable for filtering, why is it reliable for final success labels?

6. Motivation and definition of VSS are hard to follow

Figure 4 is used to motivate “developing a metric,” but VSS has not yet been defined at that point, making the figure difficult to interpret on first read.

**Questions:**

1. White-box assumptions and transferability

- Is SlotGCG a pure white-box attack (requiring attention weights) during both scoring and optimization?

- If so, can the resulting adversarial prompts transfer to other models without further optimization (zero-shot cross-model transfer)?

2. Effectiveness against deployed guardrails

Can you please evaluate SlotGCG against current guardrails (e.g., Llama Guard or similar safety classifiers/filters)?

3. Which VSS is shown in Figures 4 and 8?

In Figures 4 and 8, $\text{VSS}^{\text{final}}$ represents the VSS of which slot?

4. Random Multi-Position Insertion in Figure 5

What is the exact algorithm for **Random Multi-Position Insertion**? Why does random slot insertion without token optimization achieve faster convergence than GCG?

5. Ablation on only insertion and token allocation via VSS

Can you provide results for **VSS-based slot insertion only** (no token optimization), and compare them with **GCG-only token optimization** (no VSS-based slotting)? An ablation contrasting these two against the full SlotGCG would clarify each component’s contribution.

6. Effect of the **token budget (m)**

How does the **token budget \(m\)** affect SlotGCG’s **ASR**, convergence speed, and stability? Please include curves or tables showing performance as \(m\) varies.

---

> ### Author Response · Authors · 2025-11-21
>
> > ### **[W1] Threat model and usability boundaries**
> >
>
> > ### **[Q1] White-box assumptions and transferability**
> >
>
> Thank you for the detailed comment regarding transferability. To investigate this aspect, we extended the Universal Prompt Optimization procedure from GCG [1] and designed a surrogate method applicable to SlotGCG. We applied this **surrogate transferability algorithm** in our experiments, and the **complete procedure is provided in Appendix K.**
>
> The Universal Prompt Optimization procedure in SlotGCG is as follows.
>
> I. **Universal Vulnerable Slot Positions.**
>
> We use the **AggregationSlots** algorithm to locate slots that are **consistently vulnerable across behaviors** by aligning and integrating **per behavior scores into a unified vulnerability representation.**
>
> We begin by computing VSS for each behavior independently. Since behaviors $x^{(j)}$ have different lengths, **each per behavior score sequence $\mathrm{VSS}^{(j)}_s$ is linearly interpolated** onto a shared global coordinate system. This global system is defined over the universal slot set
>
> $$
> S^{\mathrm{univ}} = \lbrace 0,\dots,L_{\max} \rbrace,
> $$
>
> where $⁡L_{\max}$ is the maximum prompt length across all behaviors.
> After interpolation, the scores from all behaviors are averaged position-wise. This produces the **universal Vulnerable Slot Score,**
> $\mathrm{VSS}^{\mathrm{univ}}_{s},$
> representing slot positions that are consistently vulnerable across the entire behavior set.
>
> II. **Mapping Universal Slots to Individual Behaviors.**
>
> To apply universal slot positions to any given behavior, we use the **AttackInput** algorithm. Each universal slot index is rescaled to match the actual length $L_j$ of behavior $j$. The behavior-specific slot position is computed as
>
> $$
> s_t^{(j)} = \mathrm{round}\left(s_t^{\mathrm{univ}} \cdot \frac{L_j}{L_{\max}}\right).
> $$
>
> These mapped positions determine where adversarial tokens would be inserted for behavior $j$. The constructed input,
>
> $$
> \mathcal{I} \big(x^{(j)}, a_{1:\ell}, S^{(j)}_A\big),
> $$
>
> preserves the overarching universal slot structure while adapting it to prompts of differing lengths. This enables consistent application across heterogeneous behaviors.
>
> III. **Universal SlotGCG Optimization.**
>
> We integrate this universal slot mechanism into the universal prompt optimization algorithm introduced in GCG [1]. As new behaviors are added to the optimization set, we recompute $\mathrm{VSS}^{\mathrm{univ}}_{s}$ using the **AggregationSlots** algorithm. The updated universal slots, together with the AttackInput algorithm, are then used to construct the adversarial inputs for each prompt. This provides a unified slot selection strategy that generalizes across prompts with varying lengths and structures. **A full description of the procedure and the corresponding algorithms is provided in Appendix K.**
>
> > **[A for W1, Q1] While SlotGCG is inherently a pure white-box attack, we devised a surrogate attack method that enables its application in black-box settings.**
> >

---

> ### Author Response · Authors · 2025-11-21
>
> > ### **[W2-1,3] Transferability is underexplored**
> >
>
> > ### **[W3] Recency of attack targets**
> >
>
> We optimize the universal adversarial tokens and the universal VSS on Vicuna-7B using the 50 AdvBench behaviors from Sec. 5 over 500 optimization steps. For the 388 transfer behaviors introduced in [1], we constructed attack prompts by inserting **universal adversarial tokens into these universal slots** **using the AttackInput** algorithm. We then evaluated transfer performance on GPT-3.5-turbo, GPT-4o, Gemini 2.0 Flash, Gemini 2.5 Pro, and Vicuna-7B-1.5v. For ASR evaluation, we used only the **GPT-based judge (excluding human evaluation)**, and we denote these scores as $ASR_{GPT}$. In contrast, **manually filtered results** are referred to as $ASR_{Human}$. The revised version **will report $ASR_{Human}$ and $ASR_{GPT}$ separately.**
>
> **Table A: Transfer $ASR_{GPT}$ on 388 Harmful Behaviors**
>
> | Method | GPT-3.5-turbo | GPT-4o | Gemini 2.0 Flash | Gemini 2.5 Pro | Vicuna 7B 1.5v |
> | --- | --- | --- | --- | --- | --- |
> | **GCG** | 3.09% | 0.00% | 1.29% | 0.00% | 70.10% |
> | **GCG + Ours** | 50.77% | 1.80% | 2.06% | 3.61% | 63.40% |
> | **AttnGCG** | 25.52% | 0.52% | 0.26% | 0.26% | 64.43% |
> | **AttnGCG + Ours** | 43.04% | 1.55% | 0.00% | 0.00% | 76.80% |
> | **I-GCG** | 12.89% | 0.00% | 0.00% | 0.52% | 81.19% |
> | **I-GCG + Ours** | 40.46% | 0.77% | 0.77% | 0.26% | 79.90% |
> | **GCG-hij** | 37.89% | 0.00% | 0.00% | 1.55% | 63.66% |
> | **GCG-hij + Ours** | 41.49% | 0.26% | 4.12% | 6.70% | 61.34% |
>
> From the results above, we examined the transferability of SlotGCG and found that our method **exhibits context robustness, as illustrated by the Vicuna results.** Furthermore, under **cross-model transfer,** **SlotGCG consistently improved** the **transferability** of almost all baseline methods across all evaluated models, with **the average ASR rising from 5.24% to 12.35%.**
>
> Thank you for the opportunity to further improve our work regarding transferability. **The full details and methodology have been added to Appendix K**.
>
> > **[A for W2-1,3]** We explored transferability and observed that **SlotGCG is robust to both context variation** and **under cross-model transfer**, and it further **strengthens the transferability** of nearly all existing GCG-based attacks.
> >
>
> > **[A for W3]** We **additionally examined slot vulnerability on several recent models,** including **GPT-3.5-turbo, GPT-4o, Gemini 2.0 Flash, and Gemini 2.5 Pro**, and found that slot vulnerability **continues to facilitate jailbreak effectiveness** even on these newer systems.
> >
> ---
> > ### **[W2-2] Seed sensitivity**
> >
> Thank you for your comment. We evaluated seed sensitivity by running additional experiments with four different random seeds. The results are broadly consistent with those reported in the main paper.
>
> Importantly, t**he VSS-based token allocation step is determined by attention patterns** and therefore **does not depend on random seeds.** Any variation across different seeds arises entirely from the stochastic components of the underlying GCG optimization. The result presented in the paper was obtained using a widely used setting, with a random seed of 42.
>
> **Table B: Seed Sensitivity Results for SlotGCG**
>
> | Seed | $ASR_{Human}$ | $ASR_{GPT}$ |
> | --- | --- | --- |
> | **0** | 82.00% | 98.00% |
> | **1** | 70.00% | 94.00% |
> | **2** | 86.00% | 96.00% |
> | **3** | 84.00% | 94.00% |
> | **42 (Ours)** | 82.00% | 96.00% |
>
> **Table C: Seed Sensitivity Results for GCG**
>
> | Seed | $ASR_{Human}$ | $ASR_{GPT}$ |
> | --- | --- | --- |
> | **0** | 44.00% | 76.00% |
> | **1** | 46.00% | 78.00% |
> | **2** | 68.00% | 82.00% |
> | **3** | 62.00% | 74.00% |
> | **42 (Ours)** | 52.00% | 66.00% |
>
> > **[A for W2-2]** **SlotGCG exhibits low seed sensitivity**, as VSS-based token allocation is deterministic; all variability across different seeds arises from GCG’s inherent stochasticity.
> >
> ---
> **References**
>
> [1] Zou, Andy, et al. "Universal and transferable adversarial attacks on aligned language models." *arXiv preprint arXiv:2307.15043* (2023).

---

> ### Author Response · Authors · 2025-11-21
>
> > ### **[W4] Hyperparameter choices lack justification**
> >
>
> Thank you for your insightful comment on our justification of the hyperparameter choices. We agree that the explanation for the choices of the VSS temperature, the selection of upper-half layers, and the use of after-chat template tokens was insufficient. **We will add the corresponding analyses and experimental results in the appendix.**
>
> I. **Effect of Temperature $T$ on ASR Result.**
>
> We evaluated the **effect of the temperature by comparing $T$ = 1, 4, 8, 16, 32, 64**. When $T$ is too small, the distribution over slots becomes overly concentrated, leading to sub-optimal slot selection. Conversely, when **$T$ is too high,** the probability spreads out too much, which **disrupts important tokens** and often causes the **model’s responses to go off-topic**.
>
> We also observed that $T$ = 4 performs reasonably well under the default setting. In contrast, its **$ASR_{GPT}$ decreases** by an average of **45.5%** when **defenses are applied.** However, when **$T$ = 8**, its **$ASR_{GPT}$ drops only** by an average of **30.5%.**
>
> Overall, **$T$ = 8 provided the most stable performance across settings.**
>
> **Table D: ASR Variation across Temperature Settings**
>
> | temperature $T$ | $ASR_{Human}$ | $ASR_{GPT}$ |
> | --- | --- | --- |
> | **1** | 60.00% | 90.00% |
> | **4** | 80.00% | 96.00% |
> | **8 (Ours)** | 82.00% | 96.00% |
> | **16** | 74.00% | 96.00% |
> | **32** | 72.00% | 98.00% |
> | **64** | 78.00% | 96.00% |
>
> II. **Upper-half Layers and After-chat Template Tokens.**
>
> In our study, **we define the upper-half layers** as the set $\mathcal{L}_{UH} = { \lfloor L/2 \rfloor, \ldots, L }$, as described in line 262. **This corresponds to the top 50% of the decoder layers in the transformer architecture.** We acknowledge that the definition was insufficiently explained, and we will clarify this more thoroughly in the revised version.
>
> Our choices for **layer selection and the keys each query attends to are grounded in prior jailbreak analyses** [2, 3]. These studies show that attention related to jailbreaks mainly occurs in the upper layers and that after-chat template tokens serve as stable anchors from which adversarial tokens receive strong attention.
>
> Based on these observations, SlotGCG computes VSS using attention only from the upper-half layers and focuses specifically on attention originating from the after-chat template tokens.
>
> To validate this design, we **compared upper-half layers with lower-half and randomly selected layers**. Upper-half layers consistently yielded the highest ASR, supporting this design choice.
>
> **Table E: ASR Comparison across Layer Selection Strategies**
>
> | layer $L$ | $ASR_{Human}$ | $ASR_{GPT}$ |
> | --- | --- | --- |
> | **random** | 78.00% | 96.00% |
> | **lower-half** | 80.00% | 94.00% |
> | **upper-half (Ours)** | 82.00% | 96.00% |
>
> Overall, upper-half layers show a consistent advantage, achieving **the best $ASR_{GPT}$ at 96%**, slightly higher than both random and lower-half selections. Consistent with prior findings [2, 3], this indicates that **higher layer attention provides more useful signals for jailbreak generation**.
>
> > **[A for W4]** The revised version **will include these analyses and experimental justifications for our hyperparameter choices**. The effectiveness of our selected layer configuration and the after-chat template has already been demonstrated in prior work.
> >
> ---
> **References**
>
> [2] Ben-Tov, Matan, Mor Geva, and Mahmood Sharif. "Universal Jailbreak Suffixes Are Strong Attention Hijackers." *arXiv preprint arXiv:2506.12880* (2025).
>
> [3] Wang, Zijun, et al. "AttnGCG: Enhancing jailbreaking attacks on LLMs with attention manipulation." *arXiv preprint arXiv:2410.09040* (2024).

---

> ### Author Response · Authors · 2025-11-21
>
> > ### **[W5] Clarifications on the Sec. 5.3 and the role of the Perplexity Filter and GPT-based judging**
> >
>
> > **[W5-1] Inconsistency between the Perplexity Filter and Erase-and-Check ASR results**
> >
>
> Thank you for your comment pointing out this inconsistency. In the Sec. 5.3, we incorrectly described the Perplexity Filter defense. It is not **Erase-and-Check** but the **Perplexity Filter** that causes the largest reduction in ASR. Because the Perplexity Filter still strongly suppresses all GCG-style attacks, our method also fails to overcome this defense and attains ASR close to zero under it. We will correct this description in the revised version.
>
> > **[W5-2] Reliability Concerns in GPT-Based Filtering and Its Relation to Final ASR Evaluation**
> >
>
> We acknowledge that our description of the relationship between GPT-4–based filtering and the final ASR may have caused confusion. For clarity, our evaluation proceeds in three stages:
>
> 1. **Keyword-based filtering**
> 2. **GPT-4 judge**, used only for early stopping during optimization
> 3. **Final manual verification by human annotators**
>
> Importantly, **we manually evaluate the outputs that the GPT-4 judge classifies as harmful in order to compute the final ASR.**
>
> To avoid this confusion and to enable easier comparison with prior work, the revised version will **report $ASR_{Human}$ and $ASR_{GPT}$ separately**. An example of the updated reporting format is shown below:
>
> **Table F: Comparison of $ASR_{Human}$ and $ASR_{GPT}$**
>
> | **Method** | LLaMA2-7B |  | LLaMA2-13B |  | LLaMA3.1-8B |  | Mistral |  | Vicuna |  | Qwen2.5 |  |
> | --- | --- | --- | --- | --- | --- | --- | --- | --- | --- | --- | --- | --- |
> |  | $ASR_{Human}$ | $ASR_{GPT}$ | $ASR_{Human}$ | $ASR_{GPT}$ | $ASR_{Human}$ | $ASR_{GPT}$ | $ASR_{Human}$ | $ASR_{GPT}$ | $ASR_{Human}$ | $ASR_{GPT}$ | $ASR_{Human}$ | $ASR_{GPT}$ |
> | **GCG** | 52.00% | 66.00% | 58.00% | 66.00% | 56.00% | 62.00% | 86.00% | 98.00% | 80.00% | 94.00% | 68.00% | 98.00% |
> | **GCG_hij** | 76.00% | 90.00% | 78.00% | 92.00% | 62.00% | 78.00% | 84.00% | 98.00% | 86.00% | 94.00% | 68.00% | 98.00% |
> | **AttnGCG** | 42.00% | 62.00% | 16.00% | 20.00% | 44.00% | 58.00% | 94.00% | 100.00% | 88.00% | 98.00% | 74.00% | 86.00% |
> | **I_GCG** | 62.00% | 88.00% | 56.00% | 60.00% | 46.00% | 60.00% | 90.00% | 100.00% | 96.00% | 96.00% | 82.00% | 100.00% |
> | **GCG + Ours** | 82.00% | 96.00% | 78.00% | 86.00% | 82.00% | 90.00% | 86.00% | 100.00% | 86.00% | 98.00% | 68.00% | 98.00% |
> | **GCG_hij + Ours** | 90.00% | 98.00% | 90.00% | 94.00% | 88.00% | 98.00% | 80.00% | 100.00% | 82.00% | 96.00% | 72.00% | 100.00% |
> | **AttnGCG + Ours** | 92.00% | 94.00% | 92.00% | 94.00% | 80.00% | 88.00% | 90.00% | 100.00% | 90.00% | 94.00% | 88.00% | 98.00% |
> | **I_GCG + Ours** | 90.00% | 98.00% | 94.00% | 96.00% | 72.00% | 92.00% | 90.00% | 100.00% | 96.00% | 98.00% | 86.00% | 96.00% |
>
> Finally, regarding the reliability concern, it is impractical in realistic attacker settings to manually inspect the hundreds of intermediate generations produced during optimization. Although recent studies [4, 5] highlight sources of error in GPT-based judges, **prior jailbreak research [3, 6, 7] widely uses GPT-4 as an intermediate classifier due to this practical constraint**. Our pipeline follows the same practice: GPT-4 is used only to assist optimization, whereas the final ASR is determined exclusively through human evaluation.
>
> > **[A for W5] We use a human judge for the final ASR to avoid errors from GPT-based classification, and the GPT-4 judge is widely used as an automatic classifier in prior jailbreak research [3, 6, 7].**
> >
> ---
> **References**
>
> [3] Wang, Zijun, et al. "AttnGCG: Enhancing jailbreaking attacks on LLMs with attention manipulation." *arXiv preprint arXiv:2410.09040* (2024).
>
> [4] Jung, Jaehun, Faeze Brahman, and Yejin Choi. "Trust or escalate: Llm judges with provable guarantees for human agreement." *arXiv preprint arXiv:2407.18370* (2024).
>
> [5] Huang, Hui, et al. "An empirical study of llm-as-a-judge for llm evaluation: Fine-tuned judge model is not a general substitute for gpt-4." *Findings of the Association for Computational Linguistics: ACL 2025*. 2025.
>
> [6] Chao, Patrick, et al. "Jailbreaking black box large language models in twenty queries." *2025 IEEE Conference on Secure and Trustworthy Machine Learning (SaTML)*. IEEE, 2025.
>
> [7] Wei, Alexander, Nika Haghtalab, and Jacob Steinhardt. "Jailbroken: How does llm safety training fail?." *Advances in Neural Information Processing Systems* 36 (2023): 80079-80110.

---

> ### Author Response · Authors · 2025-11-21
>
> > ### **[W6] Motivation and definition of VSS are hard to follow**
> >
>
> Thank you for your comment, which helps improve the clarity and readability of our paper.
>
> > **[A for W6]** We agree with the placing Figure 4 before defining VSS can cause confusion on the first read. In the revised version, we will **move the VSS definition to the beginning of Sec. 3.2** so that it appears before Figure 4.
> >
> ---
> > ### **[Q2] Effectiveness against deployed guardrails**
> >
>
> We additionally evaluated SlotGCG under a deployed guardrail by using **Llama Guard 3** as a safety classifier. In this setting, the GPT-based evaluation was performed only when Llama Guard 3 classified the model output as safe.
>
> **Table G: SlotGCG Performance Under Llama Guard 3**
>
> | Method | Llama Guard 3 |  |
> | --- | --- | --- |
> |  | $ASR_{Human}$ | $ASR_{GPT}$ |
> | **GCG** | 16.00% | 26.00% |
> | **GCG + Ours** | 16.00% | 26.00% |
> | **AttnGCG** | 10.00% | 18.00% |
> | **AttnGCG + Ours** | 12.00% | 22.00% |
> | **I-GCG** | 14.00% | 24.00% |
> | **I-GCG + Ours** | 20.00% | 22.00% |
> | **GCG-Hij** | 16.00% | 26.00% |
> | **GCG-Hij + Ours** | 24.00% | 30.00% |
>
> On average, **SlotGCG increases $ASR_{GPT}$ from 23.50% to 25.00%**. These results indicate that SlotGCG remains competitive under deployed guardrails such as Llama Guard 3. These additional defense (SmoothLLM (Insert/Patch), Robust Prompt Optimization (RPO) and SafeDecoding, Llama Guard 3) results have also been incorporated into the revised version as Table 3.
>
> > **[A for Q2]** **SlotGCG remains effective even when guardrails such as Llama Guard 3** are applied.
> >
> ---
> > ### **[Q3] Which VSS is shown in Figures 4 and 8?**
> >
>
> The VSS values in both figures come from the **Exhaustive Slot Scan** experiment. For each slot in a given prompt, we insert a 5-token adversarial sequence into that slot, generate one candidate prompt per slot, and compute both the VSS and the final optimized loss for that slot.
>
> For example, the prompt “How to make a bomb” contains 6 slots, represented as
> ”[] How [] to [] make [] a [] bomb []”; we therefore generate 6 candidate variants, each with adversarial tokens inserted at exactly one of these slots.
>
> In **Figure 4**, **each point corresponds to the VSS of a specific insertion slot**, computed after inserting adversarial tokens into that slot. The plot shows the negative correlation between per-slot VSS and the final optimized loss.
>
> **Figure 8** visualizes the same per-slot VSS values, but with **slot indices normalized to $[0,1]$**. The dashed curves show the VSS before and after optimization (yellow: before, blue: after), while the solid curve represents the final loss obtained when optimizing adversarial tokens at that slot.
>
> We also apologize for the mistake in Figure 8, where the axis labels were incorrectly swapped. In the revised version, we have corrected the axes to **x-axis: Normalized Slot** and **y-axis: VSS**.
>
> > **[A for Q3]** **Figure 4** inserts a 5-token adversarial sequence into each slot to create one candidate per slot, optimizes all candidates, and **plots the resulting per-slot VSS against the optimized loss to show their correlation**. **Figure 8 presents these per-slot VSS values** with slot indices normalized to 0–1, with the x-axis as the normalized slot and the y-axis as the VSS.
> >
> ---
> > ### **[Q4] Random Multi-Position Insertion in Figure 5**
> >
>
> First, Figure 5 does not mean that Random Multi-Position Insertion intrinsically converges faster than GCG. The comparison includes **only those prompts for which standard GCG failed to succeed but the attack succeeded after applying Random Multi-Position Insertion followed by optimization**.
>
> We agree that the phrasing “*successful random insertion*” in Line 304 was confusing, and **we will revise it accordingly with a more detailed explanation.**
>
> The exact procedure for Random Multi-Position Insertion is as follows (Sec. 3.1):
>
> 1. We begin with 20 initial adversarial tokens and randomly partition them into a set of sequences $\mathrm{A} = \{\mathrm{a}_1^{k_1}, \ldots, \mathrm{a}_m^{k_m}\}$,
>
>  where the sequence lengths satisfy $\sum_{i=1}^{m} k_i = 20$.
>
> 2. We then insert these randomly sized sequences $A$ into randomly sampled slots $\mathrm{S_A} \subseteq S$ from the full slot set $S$, and subsequently apply GCG optimization.
>
> > **[A for Q4]** Random Multi-Position Insertion does not converge faster than GCG in general; **Figure 5 shows only the cases where GCG fails but Random Multi-Position Insertion succeeds at jailbreak.**
> >

---

> ### Author Response · Authors · 2025-11-21
>
> > ### **[Q5] Ablation on only insertion and token allocation via VSS**
> >
>
> Thank you for your comment suggesting that we clarify the individual contributions of VSS-based slot allocation and token optimization. We performed an ablation comparing (i) **GCG-only suffix insertion** (step-0, no optimization) and (ii) **VSS-based slot insertion only** (step-0, no optimization).
>
> **Table H: Ablation of Suffix-Only (Step-0) vs. VSS-Only (Step-0)**
>
> | Model | **Suffix-Only** (step-0) |  | VSS-only (step-0) |  |
> | --- | --- | --- | --- | --- |
> |  | $ASR_{Human}$ | $ASR_{GPT}$ | $ASR_{Human}$ | $ASR_{GPT}$ |
> | **Llama2-7B** | 0.00% | 0.00% | 0.00% | 0.00% |
> | **Llama2-13B** | 0.00% | 0.00% | 0.00% | 0.00% |
> | **Llama3-8B** | 0.00% | 0.00% | 0.00% | 0.00% |
> | **Vicuna-7B** | 6.00% | 6.00% | 2.00% | 2.00% |
> | **Mistral-7B** | 12.00% | 12.00% | 16.00% | 16.00% |
> | **Qwen2.5-7B** | 0.00% | 0.00% | 0.00% | 0.00% |
>
> These results indicate that step-0 insertion alone, whether suffix-based or VSS-based, does not meaningfully increase ASR, making it difficult to isolate the contribution of VSS from ASR alone.
>
> **To better understand the role of VSS,** we **analyzed its impact on the LLM’s output distribution.** We inserted 20 random tokens using either (i) VSS-based slot allocation or (ii) suffix insertion as in GCG, and measured the resulting change in the model’s first-token probability distribution over 100 trials using L2 Distance, Cosine Similarity, KL Divergence, and Top-1 Change Rate.
>
> I. **VSS-based Slot Allocation Produces Substantially Larger Shifts in the Output Distribution.**
>
> We found that **VSS-based slot allocation produces substantially larger shifts** in the output distribution than suffix insertion, indicating that VSS identifies positions where adversarial tokens exert disproportionately strong influence.
>
> **Table I: LLM Output Distribution Shift from VSS-Based Slot Allocation**
>
> | Metric | VSS-based Allocation (Ours) | Suffix (GCG) |
> | --- | --- | --- |
> | **L2 Distance              (↑)** | 2101.74 | 1079.51 |
> | **Cosine Similarity      (↓)** | 0.43 | 0.85 |
> | **KL Divergence          (↑)** | 1.20 | 0.10 |
> | **Top-1 Change Rate  (↑)** | 0.30 | 0.06 |
>
> VSS-based allocation causes **substantially larger shifts in the model’s output distribution** than suffix insertion.
>
> This supports our claim that **high-VSS slots are positions where adversarial tokens have disproportionately strong influence**, explaining why SlotGCG achieves higher jailbreak success when optimization is applied on top of VSS-guided insertion.
>
> We will add a brief explanation of this effect in the appendix of the revised version.
>
> > **[A for Q5]** While simple insertion alone does not isolate component-level contributions, our distributional analysis shows that **VSS identifies slots that yield significantly larger output** perturbations, **clarifying the contribution of VSS** in SlotGCG’s effectiveness.
> >
> ---
> > ### **[Q6] Effect of the token budget $m$**
> >
>
> SlotGCG inherits the discrete optimization structure of GCG, so the effect of the token budget $m$ follows similar patterns observed in prior GCG variants.
>
> We conducted additional experiments on Llama-2-7B with **$m$ = 5, 10, 20**. A smaller token budget consistently resulted in **lower ASR** and **slower convergence**, whereas larger budgets yielded higher ASR and faster convergence.
>
> **Table J: Effect of Token Budget ($m$) on ASR**
>
> | Method | $m=5$ |  | $m=10$ |  | $m=20$ |  |
> | --- | --- | --- | --- | --- | --- | --- |
> |  | $ASR_{Human}$ | $ASR_{GPT}$ | $ASR_{Human}$ | $ASR_{GPT}$ | $ASR_{Human}$ | $ASR_{GPT}$ |
> | **GCG** | 8.00% | 20.00% | 34.00% | 58.00% | 52.00% | 66.00% |
> | **GCG + Ours** | 62.00% | 76.00% | 76.00% | 92.00% | 82.00% | 96.00% |
>
> **Table K: Effect of Token Budget ($m$) on Average Convergence Speed**
>
> | Method | $m=5$ | $m=10$ | $m=20$ |
> | --- | --- | --- | --- |
> | **GCG** | 98.80 iter | 143.00 iter | 138.10 iter |
> | **GCG + Ours** | 68.05 iter | 32.04 iter | 40.50 iter |
>
> These results show that **reducing the token budget $m$** lowers ASR and generally slows convergence, while larger budgets tend to improve both metrics.
>
> For example, the $ASR_{GPT}$ of GCG **increases** from 20% at $m$=5 to 66% at $m$=20, and **our method shows a similar trend**, rising from 76% to 96%.
>
> > **[A for Q6]** **Smaller token budgets lead to reduced ASR and slower convergence.**
> >

---

> > ### Comment · Reviewer_rRJL · 2025-11-28
> >
> > The response has addressed my questions. I will raise my score.

---

> ### Author Response · Authors · 2025-11-28
>
> Thank you for your response and for raising your rating. We're pleased that our rebuttal was able to address your concerns. Your comments will be very helpful in enhancing the paper.
>
> It appears that OpenReview is currently unable to update review scores due to the recent API security incident. We hope that you, as a reviewer, have not been affected by this issue.
> We look forward to the scores being properly updated and will wait until the system returns to normal.

---

### Official Review · Reviewer_ELoX · 2025-11-03

**Soundness:** 2
**Presentation:** 3
**Contribution:** 2
**Rating:** 4
**Confidence:** 4

**Summary:**

This paper introduces SlotGCG, a novel extension of gradient-based jailbreak optimization that explicitly models positional vulnerability in prompts. The key idea is to identify slots—token-level positions that are more susceptible to adversarial perturbation—using an attention-derived Vulnerable Slot Score (VSS). The method first probes each slot’s sensitivity, then assigns probabilistic insertion weights and integrates them into the GCG optimization loop. Experiments across multiple open-weight LLMs and defenses demonstrate that SlotGCG improves attack success rate, convergence efficiency, and robustness against defense mechanisms, while remaining lightweight and compatible with existing frameworks.

**Strengths:**

The paper connects positional token vulnerability with optimization-based jailbreaks, introducing VSS as a quantifiable measure of slot sensitivity. The slot-probing stage is lightweight and can be easily integrated into other attack pipelines, enhancing general applicability.

**Weaknesses:**

1. Tokenizer dependence – As slots are token-based, specify the tokenizer used and discuss whether different tokenizers could affect slot boundaries or results.

2. Optimality of Step 3 formula – It is unclear whether the slot-selection formula is optimal. Would selecting top-k slots and renormalizing yield different outcomes? Clarify if this is a tunable hyperparameter and analyze its effect on ASR and prompt coherence.

3. Defense and baselines – The defense side lacks diversity and novelty. The chosen baselines and target models are relatively standard and dated. Including stronger or more recent defense baselines (e.g., [1] [2]) would strengthen the experimental credibility.

4. Limited contribution – The method builds on optimization-based jailbreak attacks (e.g., GCG), yet its improvements appear easily neutralized by simple defense strategies. This raises the question of why such an optimization-based formulation is chosen in the first place. If the approach can be trivially mitigated, the paper should clarify what fundamental insight or practical benefit this “slot vulnerability” perspective contributes beyond existing optimization-based jailbreak methods.

5. Target model and PPL results – Sec. 5.3 does not specify the target model, and the statement that “PPL mitigation is moderate” seems inconsistent with near-zero results. Please clarify both.

6. Unclear notation – In Step 3 of Sec. 4, the variables fsi and S* are undefined. Add explicit notation or a brief symbol explanation for clarity.

7. Minor textual error – Line 213 should mention three prompts instead of four. Please verify and correct.

8. Slot normalization – In Sec. 3.1, slot indices are normalized by the longest prompt in the batch, which likely prevents values near 1.0. The motivation and comparison with per-prompt normalization should be clarified.

[1] Robust Prompt Optimization for Defending Language Models Against Jailbreaking Attacks

[2] SafeDecoding: Defending against Jailbreak Attacks via Safety-Aware Decoding

**Questions:**

see above

---

> ### Author Response · Authors · 2025-11-21
>
> We appreciate the thoughtful feedback on our paper, and we included additional comments to further improve our work.
>
> > ### **[W1] : Tokenizer effects on slot boundaries and results**
> >
>
> Thank you for your comment. Our study evaluated a diverse set of models, enabling clear comparisons of **tokenizer effects**. Specifically, we used Llama 2–7B, Llama 2–13B, Vicuna 7B v1.5, Llama 3.1–8B, Mistral 7B v2, and Qwen2.5–7B.
>
> **These seven models use the following three tokenizer families**
> - **Llama2-family tokenizers** : Llama 2-7B, Llama 2-13B, Vicuna 7B v1.5
> - **Llama3-family tokenizers** : Llama 3.1-8B
> - **Mistral-family tokenizers** : Mistral 7B v2
> - **Qwen2.5-family tokenizers** : Qwen2.5-7B
>
> **As the reviewer noted, slot boundaries vary depending on the tokenizer used.** As shown in Table A, the average number of slots across the 50 behaviors changes correspondingly.
>
> **Table A: Average Number of Slots by Tokenizer Family**
>
> |  | Llama2 Family | Llama3 Family | Mistral Family | Qwen2.5 Family |
> | --- | --- | --- | --- | --- |
> | **Number of slots** | 12.73 | 9.92 | 10.76 | 9.96 |
>
> **The number of slots differs by tokenizer**, indicating that **slot boundaries may change accordingly depending on the tokenizer.**
>
> However, our experimental results indicate that changes in slot boundaries have little to no impact on ASR performance.
>
> **To ensure comparability with prior work**, we report **unfiltered results as $ASR_{GPT}$**. In contrast, **manually filtered** results are referred to as **$ASR_{Human}$.**
>
> **Table B: ASR of Different Models Using Different Tokenizers**
>
> | Method | Llama2 Family |  | Llama3 Family |  | Mistral Family |  | Qwen2.5 Family |  |
> | --- | --- | --- | --- | --- | --- | --- | --- | --- |
> |  | $ASR_{Human}$ | $ASR_{GPT}$ | $ASR_{Human}$ | $ASR_{GPT}$ | $ASR_{Human}$ | $ASR_{GPT}$ | $ASR_{Human}$ | $ASR_{GPT}$ |
> | **GCG + Ours** | 82.00% | 90.00% | 82.00% | 90.00% | 86.00% | 100.00% | 68.00% | 98.00% |
> | **GCG_hij + Ours** | 87.33% | 96.00% | 88.00% | 98.00% | 80.00% | 100.00% | 72.00% | 100.00% |
> | **AttnGCG + Ours** | 91.33% | 94.00% | 80.00% | 88.00% | 90.00% | 100.00% | 88.00% | 98.00% |
> | **I_GCG + Ours** | 93.33% | 97.33% | 72.00% | 92.00% | 90.00% | 100.00% | 86.00% | 96.00% |
> | **Avg** | **88.50%** | **94.33%** | **80.50%** | **92.00%** | **86.50%** | **100.00%** | **78.50%** | **98.00%** |
>
> **Table C: ASR of Different Models Using Different Tokenizers**
>
> | Method | Llama2 Family |  | Llama3 Family |  | Mistral Family |  | Qwen2.5 Family |  |
> | --- | --- | --- | --- | --- | --- | --- | --- | --- |
> |  | $ASR_{Human}$ | $ASR_{GPT}$ | $ASR_{Human}$ | $ASR_{GPT}$ | $ASR_{Human}$ | $ASR_{GPT}$ | $ASR_{Human}$ | $ASR_{GPT}$ |
> | **GCG** | 63.33% | 75.33% | 56.00% | 62.00% | 86.00% | 98.00% | 68.00% | 98.00% |
> | **GCG_hij** | 80.00% | 92.00% | 62.00% | 78.00% | 84.00% | 98.00% | 68.00% | 98.00% |
> | **AttnGCG** | 48.67% | 60.00% | 44.00% | 58.00% | 94.00% | 100.00% | 74.00% | 86.00% |
> | **I_GCG** | 71.33% | 81.33% | 46.00% | 60.00% | 90.00% | 100.00% | 82.00% | 100.00% |
> | **Avg** | **65.83%** | **77.17%** | **52.00%** | **64.50%** | **88.50%** | **99.00%** | **73.00%** | **95.50%** |
>
> Tables B and C report the ASR results of our methods and prior methods across models using different tokenizers. Although there are slight differences in ASR, **the extent of this variation is comparable to that of tokenizer independent methods (GCG, GCG_hij, etc.)**. In particular, the average **$ASR_{GPT}$** values **range** only from **92–100%** in Table B and **64.5–99%** in Table C, showing a degree of **variability similar to that of the baseline methods.** This suggests that the **differences in ASR mainly arise from model alignment rather than tokenizer selection**, indicating that tokenizer dependence is minimal.
>
> > **[A for W1]** : As slots are token-based, **the dependence of slot boundaries on the tokenizer is unavoidable**. Nevertheless, **our methodology maintains strong performance across tokenizers**, because it **effectively identifies quantitatively vulnerable points regardless of which tokenizer is used.**
> >

---

> ### Author Response · Authors · 2025-11-21
>
> > ### **[W2] Optimality of Step 3 formula**
> >
>
> Thank you for your comment and we think the top-$k$ slots idea is a useful way to examine whether our approach is truly optimal.
>
> **The top-$k$ slots method can be applied easily to our work and the top-$k$ is a tunable hyperparameter.** We identified the top-$k$ VSS values based on their scores, renormalized them, and allocated tokens to the corresponding top-$k$ slots according to the renormalized VSS.
>
> I. **ASR results According to Top-$k$.**
>
> We applied top-$k$ slots method to GCG + Ours, using $k$ = 1, 2, 4, and 8. Since our original method uses all slots, we denote it as “inf”.
>
> **Table D**: **Effect of Top-$k$ Slot Selection on ASR Performance**
>
> | top-$k$ | $ASR_{Human}$ | $ASR_{GPT}$ |
> | --- | --- | --- |
> | **1** | 64.00% | 82.00% |
> | **2** | 72.00% | 94.00% |
> | **4** | 82.00% | 94.00% |
> | **8** | 82.00% | 98.00% |
> | **inf (Ours)** | 82.00% | 96.00% |
>
> Table D presents the change in ASR with respect to the choice of top-$k$. **We observe that selecting more slots (i.e., approaching Ours) leads to improved performance.** Our method typically selects an average of 13.82 slots, **indicating that performance tends to increase as the slot selection becomes closer to our approach.**
>
> II. **Prompt Coherence by Top-$k$.**
>
> We interpret the concern regarding “*prompt coherence*” as asking how the **structure of the generated attack prompt** changes depending on the choice of top-$k$. This can be examined through the **distribution of tokens per slot.**
>
> **Table E: Statistics of Tokens per Slot Under Different Top-$k$ Settings**
>
> | top-$k$ | Mean | Std |
> | --- | --- | --- |
> | **1** | 20.00 | 0.00 |
> | **2** | 10.00 | 0.00 |
> | **4** | 5.00 | 0.00 |
> | **8** | 2.51 | 0.05 |
> | **inf (Ours)** | 1.50 | 0.32 |
>
> As shown in **Table E**, as **$k$ increases**, the average tokens per slot decreases from **20.00 to 1.50**, indicating that the **adversarial tokens are more broadly distributed** **across the entire prompt.** In contrast, when **$k$ is larger,** the standard deviation becomes increases  from **0.00 to 0.32**, meaning that the **number of tokens inserted into each slot varies more**.
> These results suggest that **increasing $k$** leads to greater changes in **the prompt coherence.**
>
> > #### **[A for W2]** In summary, **top-$k$ serves as a tunable hyperparameter**. We observe consistent **ASR performance gains as the slot-selection strategy more closely aligns with our method**, suggesting that the **Step 3 formulas are close to optimal.** Furthermore, **increasing $k$** results in **greater changes to the prompt coherence.**
> >

---

> ### Author Response · Authors · 2025-11-21
>
> > ### **[W3] Lacks diversity and novelty defense.**
> >
>
> Thank you for your comment, which helps us make our defense evaluation more robust.
>
> Following your suggestion, we additionally evaluated the more recent defenses, **Robust Prompt Optimization (RPO)** and **SafeDecoding**. We also conducted experiments on further variants of **SmoothLLM** (Insert/Patch) and on **Llama Guard 3** [1].
>
> We used the same hyperparameters as SWAP for all variations of SmoothLLM (6 perturbed variants, $q$ = 5%). For RPO, we used the suffix provided by the original authors. SafeDecoding was also used with its default settings ($\alpha$ = 3, $m$ = 2, top-$k$ = 10, $c$ = 5). When using Llama Guard as a defense, we proceeded with the GPT Judge only when Llama Guard classified the output as safe.
>
> **Table F: Additional Defense Results**
>
> | Method | SmoothLLM (Swap) |  | SmoothLLM (Insert) |  | SmoothLLM (Patch) |  | RPO |  | SafeDecoding |  | Llama Guard 3 |  |
> | --- | --- | --- | --- | --- | --- | --- | --- | --- | --- | --- | --- | --- |
> |  | $ASR_{Human}$ | $ASR_{GPT}$ | $ASR_{Human}$ | $ASR_{GPT}$ | $ASR_{Human}$ | $ASR_{GPT}$ | $ASR_{Human}$ | $ASR_{GPT}$ | $ASR_{Human}$ | $ASR_{GPT}$ | $ASR_{Human}$ | $ASR_{GPT}$ |
> | **GCG** | 44.00% | 44.00% | 22.00% | 38.00% | 24.00% | 62.00% | 32.00% | 46.00% | 8.00% | 10.00% | 16.00% | 26.00% |
> | **GCG + Ours** | 86.00% | 96.00% | 76.00% | 92.00% | 76.00% | 96.00% | 30.00% | 46.00% | 10.00% | 12.00% | 16.00% | 26.00% |
> | **AttnGCG** | 30.00% | 32.00% | 18.00% | 32.00% | 28.00% | 48.00% | 34.00% | 52.00% | 6.00% | 8.00% | 10.00% | 18.00% |
> | **AttnGCG + Ours** | 92.00% | 94.00% | 72.00% | 94.00% | 72.00% | 96.00% | 44.00% | 64.00% | 20.00% | 26.00% | 12.00% | 22.00% |
> | **GCG-Hij** | 44.00% | 44.00% | 32.00% | 54.00% | 52.00% | 78.00% | 42.00% | 54.00% | 8.00% | 8.00% | 16.00% | 26.00% |
> | **GCG-Hij + Ours** | 96.00% | 96.00% | 66.00% | 94.00% | 64.00% | 98.00% | 38.00% | 66.00% | 26.00% | 28.00% | 24.00% | 30.00% |
> | **I-GCG** | 44.00% | 48.00% | 28.00% | 44.00% | 36.00% | 64.00% | 36.00% | 60.00% | 14.00% | 14.00% | 14.00% | 24.00% |
> | **I-GCG + Ours** | 96.00% | 98.00% | 82.00% | 98.00% | 80.00% | 98.00% | 38.00% | 60.00% | 18.00% | 20.00% | 20.00% | 22.00% |
>
> Across all these defenses, **SlotGCG consistently outperforms the baseline attack, raising the overall average $ASR_{GPT}$ from 38.91% to 65.50%.** This improvement shows that its robustness does not stem from exploiting a single defense’s weakness, but rather from its ability to distribute adversarial tokens across multiple high VSS insertion slots.
>
> These results also show that SlotGCG achieves higher ASR than the baseline attacks, demonstrating that the attack **remains effective even under stronger and more recent defense mechanisms. These additional defense results have also been incorporated into the revised version as Table 3.**
>
> > **[A for W3]** **SlotGCG continues to outperform** baseline methods **under stronger or more recent defenses** (RPO, SafeDecoding, Llama Guard 3, and extended SmoothLLM variants).
> >
> ---
> **References**
>
> [1] Dubey, Abhimanyu, et al. "The llama 3 herd of models." *arXiv e-prints* (2024): arXiv-2407.

---

> ### Author Response · Authors · 2025-11-21
>
> > ### **[W4] Limited contribution**
> >
>
> Thank you for your comment, which prompted us to reconsider the fundamental aspects of our method.
>
> I. **Optimization-based Methods Offer the Following Advantages.**
>
> Optimization-based jailbreak attacks **use direct model information** such as **gradients** and **logits**, and explicitly optimize the model toward producing harmful outputs. This allows us to understand **how specific inserted tokens or local noise** strongly **influence the output distribution**.
>
> Such **interpretability is important** from a **red-teaming** perspective because it reveals why the model becomes vulnerable rather than only showing that the attack succeeds.
>
> II. **Understanding Slot Vulnerability In the Context of the Model’s Output Distribution.**
>
> We conducted experiments to **analyze slot vulnerability** from the **perspective of its impact on the output distribution**. We allocated **20 random tokens according to VSS** and **compared the output distribution shift** with that of simply **appending** the same tokens at the **suffix position** used in GCG. In both cases, we measured the shift relative to the original input without any token insertion and then compared the two shifts.
>
> We measured the L2 Distance, Cosine Similarity, KL Divergence, and Top-1 Change Rate of the first-token output distribution of Llama-2 over 100 trials. **High L2 Distance, KL Divergence**, and **Top-1 Change Rate**, as well as **low Cosine Similarity,** indicate that the token insertion **induces a large shift in the output distribution.**
>
> **Table G. Change in LLM output distribution (Llama2 7b)**
>
> | Metric | VSS-based Allocation (Ours) | Suffix (GCG) |
> | --- | --- | --- |
> | **L2 Distance              (↑)** | 2101.74 | 1079.51 |
> | **Cosine Similarity      (↓)** | 0.43 | 0.85 |
> | **KL Divergence          (↑)** | 1.20 | 0.10 |
> | **Top-1 Change Rate  (↑)** | 0.30 | 0.06 |
>
> **Table G** presents that **allocating tokens based on VSS induces** substantially **larger perturbations to the output distribution** than simply appending them at the suffix. This confirms that **slot vulnerability identifies positions that strongly affect the LLM’s output distribution** and explains why our method is more effective than GCG in jailbreak settings.
>
> We further expect that this positional vulnerability perspective may extend to related tasks such as topic steering and controlled generation.
>
> We thank your comment again for helping us articulate this insight more clearly.
>
> > **[A for W4]** **Slot vulnerability corresponds** to **positions** that **induce large shifts** in the LLM’s **output distribution**, and this provides a fundamental insight into model behavior.
> >
> ---
> > ### **[W5] Clarifications on the Target Model and PPL Results in Sec. 5.3**
> >
>
> Thank you for your comment, which helps us clarify our paper.
>
> > **[W5-1] Target model clarification**
> >
>
> We acknowledge that Sec. 5.3 did not explicitly specify the target model used in the defense evaluation.
>
> All defense experiments in Sec. 5.3 were conducted on **Llama-2-7B-Chat**. We selected this model for consistency with prior jailbreak and defense studies, where it is commonly used as a standard baseline.
>
> We will explicitly state the target model in the revised version to avoid confusion.
>
> > **[W5-2] PPL results**
> >
>
> We agree that the phrase *“mitigation is moderate”* does not accurately reflect the results.
>
> In practice, the **Perplexity Filter** blocks almost all GCG-based attacks, and SlotGCG also yields ASR values close to zero under this defense.
>
> We will correct the text in the revised version to more accurately describe the strength of the Perplexity Filter.
>
> > **[A for W5] We will explicitly specify the target model in Sec. 5.3** and **revise the description of the PPL results** to accurately reflect the near-zero ASR under the Perplexity Filter.
> >
> ---
> > ### **[W6] Unclear notation**
> >
>
> Thank you for pointing this out. In Step 3 of Sec. 4, the meanings of $f_s$ and $S^*$ were not explicitly defined, which may have caused confusion.
>
> In our implementation, we compute:
>
> $r_s = m \cdot \pi_s$, $t_s = \lfloor r_s \rfloor$, $f_s = r_s - t_s$
>
> - **$f_s$** denotes the **fractional remainder** of $r_s$ after subtracting its integer part for each slot.
> - **$S^{\star}$** denotes the **top-$\bigl(m - \sum_{s_i \in S} t_{s_i}\bigr)$ slots with the largest $f_s$ values**, to which the remaining tokens are assigned.
>
> > **[A for W6]** We have **added explicit definitions of both symbols in Sec. 4** to ensure clarity in Step 3.
> >

---

> ### Author Response · Authors · 2025-11-21
>
> > ### **[W7] Minor textual error**
> >
>
> Thank you for your detailed comment and careful reading of our work and for helping us correct this ambiguity.
>
> > **[A for W7]** As the reviewer correctly pointed out, the statement in Line 213 should refer to three prompts. We have revised the text by **replacing “*four individual prompts*” with “*three individual prompts*” in the revised version**.
> >
> ---
> > ### **[W8] Slot normalization**
> >
>
> Thank you for your comment, which allows us to clarify the motivation and details of slot normalization.
>
> Slot normalization is applied **within each individual prompt**, not across a batch. For a given prompt of length $L$, we simply normalize slot indices from $0$ to $L$ into the range $[0, 1]$. **This normalization is motivated solely by the need for visualization**, enabling consistent comparison of slot positions across prompts with different lengths.
>
> The use of $L_{\text{max}}$ in the text was misleading, and we will correct it to reflect the actual implementation, where normalization is performed with respect to each prompt’s own length $L$.
>
> Importantly, this **normalization is not used when determining insertion positions**. SlotGCG always inserts adversarial tokens according to the unnormalized slot indices. We will clarify this distinction in the revised version.
>
> > **[A for W8]** **Slot normalization is performed within each prompt** and is **used solely for visualization purposes,** with the **motivation of enabling fair visual comparison** across prompts of different lengths.
> >

---

> > ### Comment · Reviewer_ELoX · 2025-11-27
> > **Thanks for the rebuttal**
> >
> > I am satisfied with most of the answers. I am updating my score. Thanks.

---

> > > ### Author Response · Authors · 2025-11-27
> > >
> > > Thank you for your response and for raising your rating. We're glad that our rebuttal helped clarify your concerns. Your insights will be valuable in improving our paper.

---

### Comment · Area_Chair_jSc8 · 2025-11-25

Dear Reviewers,

The authors have submitted their responses to your questions and feedbacks. Please read them and give your comments.

Regards, AC

---

### Author Response · Authors · 2025-12-02
**Rebuttal Summary for AC**

Dear AC,

We sincerely appreciate your time and effort in reviewing our paper. The reviewers provided highly thoughtful and constructive feedback, and we addressed their concerns thoroughly. Three reviewers indicated that they would raise their scores and did so (ELoX: 4→6, rRJL: 4→6, 99aE: 6→8), and dH2T maintained their positive score (6).

Below is a **concise summary of the key points discussed** during the rebuttal period.

---
## **Strengths**
- **Novelty : Reviewers rRJL, 99aE, and dH2T** highlighted our **novel framing of positional vulnerability**, filling a gap beyond suffix-only attacks.
- **Efficiency** : **All reviewers** emphasized the method’s **generality, lightweight design, and easy integration** into gradient-based approaches.
- **Experimental Coverage : Reviewers rRJL and dH2T** noted the **broad experimental coverage**.
---
## **Weaknesses**
> ### 1. Clarifying the Contribution of VSS in SlotGCG’s Effectiveness (ELoX, rRJL)
>

The reviewer noted that the contribution of VSS was not clearly articulated. To address this, we **compare random token** insertions at **VSS-selected positions** with insertions at **suffix positions.** Our results show that VSS guided insertions induce substantially **stronger perturbations in the LLM’s output distribution.**

> **[Rebuttal to W1]** This response **clarified how VSS contributes to SlotGCG**’s effectiveness, thereby successfully addressed the reviewers’ concerns.
>

---

> ### 2. Hyperparameter Setting (rRJL, 99aE, dH2T)
>

The reviewers noted that **our justification for the chosen hyperparameters lacked clarity.** In response, we conducted **ablation studies** on Llama-2-7B for both GCG and GCG + Ours, examining the impact of **VSS temperature, layer selection, random seed, and token budget**.

> **[Rebuttal to W2]** In all cases, we provided **clear justifications for our chosen hyperparameters** and demonstrated that our method **consistently and robustly outperforms the baseline** across different **random seeds** and **token budgets**. These responses effectively addressed the concerns raised by all three reviewers.
>

---

> ### 3. Transferability (rRJL, 99aE)
>

In response to the reviewers’ comments regarding the **absence of transferability experiments** to black-box models, we **extended GCG’s Universal Prompt Optimization algorithm** by **incorporating interpolation** to obtain a **Universal VSS** and proposed a **Universal SlotGCG algorithm (Appendix K).**

We **evaluated SlotGCG’s transferability** on **five black-box models** (GPT-3.5-turbo, GPT-4o, Gemini 2.0 Flash, Gemini 2.5 Pro, and Vicuna-7B-1.5v) using **388 harmful behaviors** that were **unseen during training.**

> **[Rebuttal to W3]** This response **clarified how our method remains effective** in the **transfer setting**, where the average **ASR increased from 5.24% to 12.35%**, thereby successfully addressing the reviewers’ concerns.
>

---

> ### 4. Additional Defenses, Attack, and Datasets (ELoX, rRJL, 99aE)
>

**Reviewers ELoX and 99aE** pointed out that our defense evaluation lacked recent strong defenses. **Reviewer rRJL** also asked about SlotGCG’s performance against classifier-based defenses such as **Llama Guard**.

To address these concerns, we conducted additional experiments using **two variants of SmoothLLM (Insert, Patch)**, as well as **RPO**, **SafeDecoding**, and **Llama Guard 3**. Across all defenses, **Ours** consistently **improved** the average **ASR from 38.91% to 65.50%.**

In addition, we evaluated our method against **GBDA**, a non-GCG attack, and found that incorporating Ours increased the average **ASR from 21.00% to 61.67%.**

We further expanded our evaluation to more recent datasets, including **JailbreakBench** and **HarmBench**, and found that **our method increased ASR by 14–18 percentage points.**

> **[Rebuttal to W4]** We additionally evaluated **one non-GCG attack, five defense methods, and two recent datasets**, which further reinforced the robustness of our findings. These responses effectively addressed the concerns raised by all three reviewers.
>
---
## **Conclusion**
- We **clarified the role of VSS in SlotGCG** by showing that VSS-guided insertions produce substantially stronger perturbations than suffix-based insertions.
- We **justified our hyperparameter choices through ablations studies** and showed that our method consistently outperforms the baseline across diverse settings.
- To address concerns about transferability, we developed **Universal SlotGCG** and observed significant **improvements in ASR across five black-box models.**
- We further **expanded our evaluation** by **incorporating additional defenses, attacks, and datasets,** thereby confirming the robustness of our method.
- All four reviewers indicated that they were satisfied with our responses. **Three reviewers stated that they would raise their scores (ELoX: 4→6, rRJL: 4→6, 99aE: 6→8), and one reviewer maintained a positive score (dH2T : 6).**

---

### Meta-Review · Area_Chair_iyPw · 2026-01-06

**Summary:**

This paper introduces SlotGCG, a method that improves upon existing gradient-based jailbreak attacks by identifying and exploiting positionally vulnerable slots within prompts. The authors propose a Vulnerable Slot Score (VSS) to quantify positional vulnerability and integrate it into optimization-based attacks, achieving higher Attack Success Rates (ASR), faster convergence, and better robustness against defenses.

The authors submitted a detailed and comprehensive rebuttal, addressing all major concerns raised by the reviewers. They conducted additional experiments, provided clarifications, and extended evaluations to strengthen the paper's contributions.

**Reviewer Concerns:**

## Clarification of VSS Contribution (ELoX, rRJL)

Concern: The role of VSS in SlotGCG’s effectiveness was unclear.

Rebuttal: Authors showed that VSS-guided token insertions produce stronger perturbations than suffix-based insertions, clarifying its contribution.

## Hyperparameter Justification (rRJL, 99aE, dH2T)

Concern: Lack of clarity in hyperparameter choices.

Rebuttal: Ablation studies on temperature, layer selection, seed, and token budget demonstrated robustness and justified selections.

## Transferability to Black-Box Models (rRJL, 99aE)

Concern: No evaluation on black-box models.

Rebuttal: Introduced Universal SlotGCG, tested on 5 black-box models (GPT-3.5/4o, Gemini, Vicuna), improving ASR from 5.24% to 12.35%.

## Additional Defenses, Attacks, and Datasets (ELoX, rRJL, 99aE)

Concern: Limited defense/attack/dataset evaluation.

Rebuttal: Added evaluations on SmoothLLM variants, RPO, SafeDecoding, Llama Guard 3, GBDA, and datasets JailbreakBench & HarmBench, consistently showing improved ASR.

## Tokenizer Dependence & Slot Boundaries (ELoX)

Concern: Tokenizer effects on slots and results.

Rebuttal: Showed consistent performance across tokenizer families (Llama2/3, Mistral, Qwen), indicating minimal impact.

**Reviewer Scores:**

Three reviewers (ELoX, rRJL, 99aE) explicitly stated they would raise their scores post-rebuttal.

One reviewer (dH2T) maintained a positive score.

The overall trajectory is strongly positive, with final scores ranging from 6 to 8.

---

### Decision · Program_Chairs · 2026-01-26

Accept (Poster)